# (p)ppGpp-mediated GTP homeostasis ensures survival and antibiotic tolerance of *Staphylococcus aureus*

Andrea Salzer [1], Sophia Ingrassia[1], Parvati Iyer[1], Lisa Sauer[1], Johanna Rapp [1], Ronja Dobritz[1], Jennifer Müller [1], Hannes Link[1,2] & Christiane Wolz [1,2] ✉

Antibiotic tolerance in non-growing bacterial populations is of major concern regarding antibiotic treatment failures. Whether and how the messenger molecule (p)ppGpp contributes to this phenomenon is controversial. We show for *Staphylococcus aureus* that (p)ppGpp-dependent restriction of the GTP pool is essential for the culturability of starved cells. Survival was independent of the GTP-responsive regulator CodY. Elevated GTP levels in a starved (p)ppGpp-deficient mutant led to quiescent state characterised by alterations in membrane architecture and a decrease of the proton motive force (PMF). This was accompanied by dysregulation of components involved in electron transport, including *qoxABCD*, encoding the main terminal oxidase. Increasing *qoxABCD* transcription by mutation of the transcription start site (iATP to iGTP) partially restored the culturability of the (p)ppGpp-deficient mutant. Thus, regulation of nucleotide-dependent promoters by altered nucleotide levels contribute to starvation adaptability. Loss of PMF under high GTP conditions also renders bacteria susceptible to antibiotics. Thus, targeting the PMF or nucleotide availability may be a valuable strategy to combat antibiotic tolerance.

Bacteria need to switch between rapid growth under favorable conditions and a nongrowing but stress-tolerant state under nutrient-limited conditions. In pathogenic bacteria, such a nongrowing state is often associated with persistent infections and antibiotic tolerance[1–3]. Nongrowing bacteria continue to synthesize macromolecules, have a basal metabolism and are prepared to resume growth[4]. Recent studies have demonstrated that the messenger molecule (p)ppGpp is central to coordinating optimal resource allocation under nutrient-limited conditions[5]. For *Escherichia coli*, (p)ppGpp was postulated to act as the major regulator of growth rate control[6,7]. Even subtle fluctuations in levels of (p)ppGpp or basal levels that might be below the detection limit, such as during entry into the stationary growth phase, can modulate cellular processes[8]. (p)ppGpp specifically interferes with replication, translation, and transcription[9]. In proteobacteria, (p)ppGpp directly binds to RNA polymerase to regulate transcription. In Firmicutes (Bacillota), however, (p)ppGpp does not interfere with RNA polymerase activity[10]. Nevertheless, (p)ppGpp leads to profound reprogramming of transcription[11]. In these organisms, (p)ppGpp-mediated transcriptional regulation is likely indirect via accompanied changes in the GTP level[12–15]. (p)ppGpp was shown to inhibit major enzymes involved in GTP synthesis and homeostasis including the guanylate kinase Gmk, the

IMP dehydrogenase GuaB, the hypoxanthine-guanine phosphoribosyl-transferase HrpT, and the transcriptional regulator PurR[16]. Thus, (p)ppGpp synthesis correlates inversely with the GTP pool. A decrease in GTP levels during the stringent response may affect transcription via the GTP-sensitive transcriptional regulator CodY[16] or through regulation of initiation nucleotide (iNTP)-sensitive promoters[12,15,17].

The role of (p)ppGpp in stationary phase survival and bacterial tolerance in *Staphylococcus aureus* is controversial[3,18–20]. In this human pathogen and other Firmicutes (Bacillota), synthesis and degradation of the messenger molecule (p)ppGpp are orchestrated by the bifunctional Rel enzyme with synthetase and hydrolase activity and by the two synthetases SasA/RelP and SasB/RelQ[9,16]. Induction of the stringent response via amino acid limitation (Rel) or cell wall stress (RelP or RelQ) contributes to pathogenicity and antibiotic tolerance[13,20]. However, the role of basal levels of (p)ppGpp in bacterial survival in the stationary phase under non-stressed conditions is less clear. Here, we show that under aerobic condition (p)ppGpp is essential for maintaining GTP homeostasis in stationary phase and thereby preserving electron transport chain activity and the proton motive force (PMF), which ensures culturability and antibiotic tolerance.

[1]Interfaculty Institute of Microbiology and Infection Medicine, Tübingen, Germany. [2]Cluster of Excellence EXC 2124 "Controlling Microbes to Fight Infections", University of Tübingen, Tübingen, Germany. ✉e-mail: christiane.wolz@uni-tuebingen.de

## Results

### (p)ppGpp contributes to culturability in the stationary phase

To study the requirement of (p)ppGpp synthesis for stationary phase survival, we monitored the growth and culturability (determined by CFU enumeration) of *S. aureus* wildtype strain HG001 and its isogenic (p)ppGpp[0] mutant in chemically defined medium (CDM) for 24 h. (p)ppGpp[0] is unable to synthesize (p)ppGpp due to mutations in all three (p)ppGpp synthetases (full deletion of *rel*, synthetase mutations in *relP* and *relQ*)[21]. There was no significant difference in growth measured by optical density ($OD_{600}$) between wild type and (p)ppGpp[0] mutant (Fig. 1A). However, after entry into the stationary phase, the number of colony-forming units (CFU) started to decrease only in the mutant. Compared to the wildtype an approximately one log reduction in CFU was observed after 24 h (Fig. 1B & S1A). Microscopical counting confirmed that the bacterial number was not different between both strains supporting that the culturability is indeed decreased in the (p)ppGpp[0] mutant (Fig. 1C).

To confirm that a typical stringent response is induced in stationary phase under our growth condition, we monitored expression levels of the well-characterized (p)ppGpp activated genes *psmα* and *rsaD*[11,22]. After entering the stationary phase (7 h), both genes are significantly higher expressed in the wildtype than in the (p)ppGpp[0] mutant (Fig. S1C).

We next confirmed that reduced culturability in (p)ppGpp[0] mutants is a general phenomenon. (p)ppGpp[0] mutants of strain USA300 and SH1000 showed a similar reduction in culturability (Fig. 1D,). USA300 is a highly virulent methicillin-resistant strain. SH1000 is a phage-cured strain closely related to HG001. Thus, decreased culturability when (p)ppGpp is lacking is independent of methicillin-resistance or native phages.

### Rel and RelQ contribute to culturability in the stationary phase

In *S. aureus* (p)ppGpp can be synthesized by three different synthetases (Rel, RelP or RelQ). Rel-dependent stringent response can be initiated by mupirocin, a tRNA synthetase inhibitor mimicking amino acid starvation. Addition of mupirocin to exponentially growing bacteria resulted in a slight

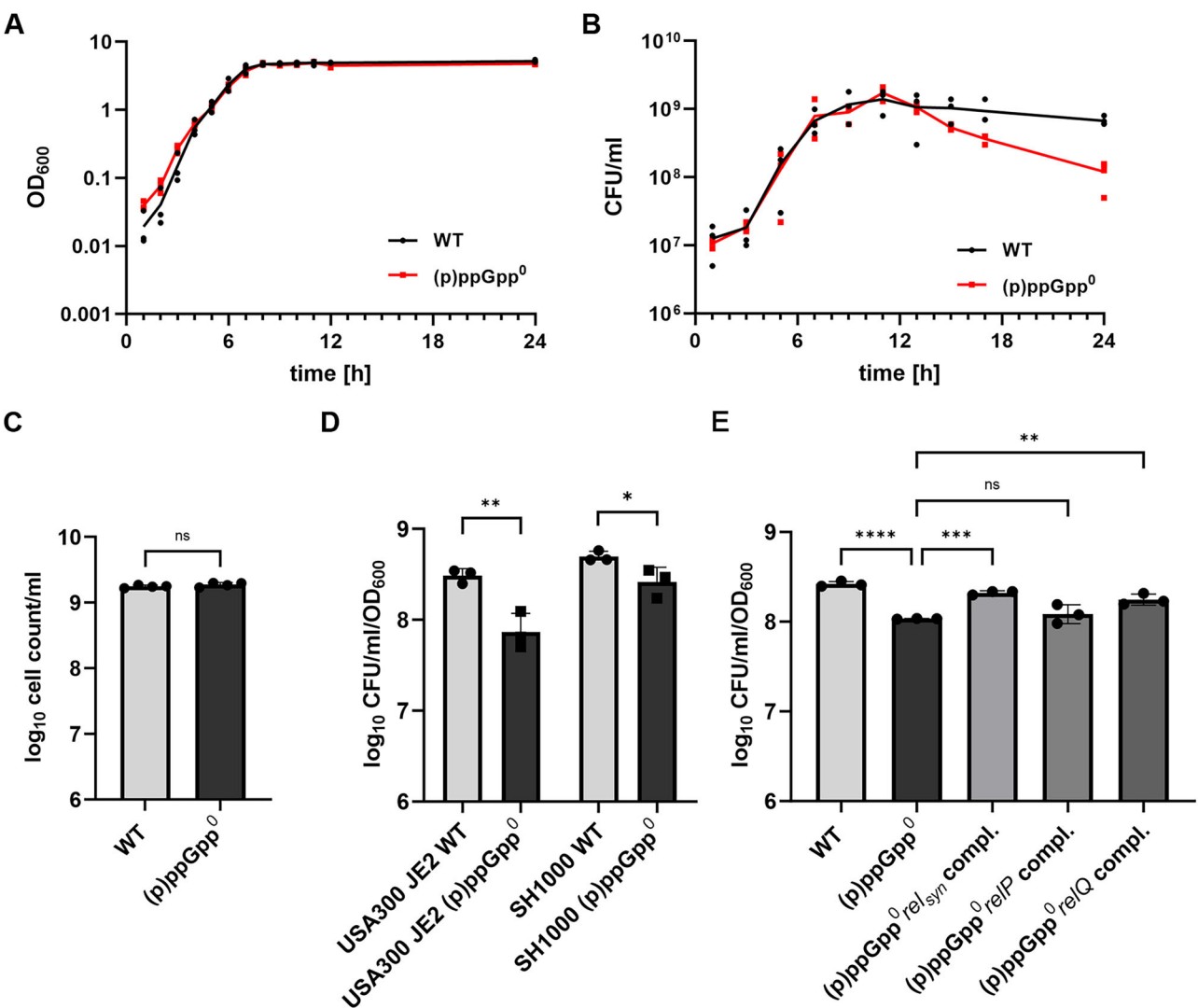

**Fig. 1 | Culturability of the (p)ppGpp[0] mutant decreases during the stationary growth phase. A** *S. aureus* HG001 wildtype (WT) and the isogenic (p)ppGpp[0] strain were cultured in CDM and growth was monitored by optical density $OD_{600}$ and (**B**) CFU/ml determination (*n* = 3). **C** *S. aureus* HG001 wildtype (WT) and the isogenic (p)ppGpp[0] strain were cultured in CDM for 24 h and bacterial cells were counted microscopically (*n* = 3, unpaired *t* test, *p* = 0.0757). **D** USA300 JE2 and SH1000 and their isogenic (p)ppGpp[0] mutants were grown in CDM for 24 h and culturability was determined by CFU/ml enumeration normalized to $OD_{600}$. Data are shown as mean ± SD (*n* = 3). Statistical significance was determined by two-tailed unpaired *t* tests performed on log₁₀ transformed data (**\**p*-value = 0.0076, *\**p*-value = 0.048). **E** Strains were grown for 24 h in CDM and culturability was determined by CFU/ml enumeration normalized to $OD_{600}$. Data shown are mean ± SD (*n* = 3). Statistical significance was determined by one-way analysis of variance (ANOVA) with a Šidák's post-test on log₁₀ transformed data (**\**\**\**p*-value < 0.0001, **\**\**p*-value = 0.0003, **\**p*-value = 0.01, ns: *p* = .09744).

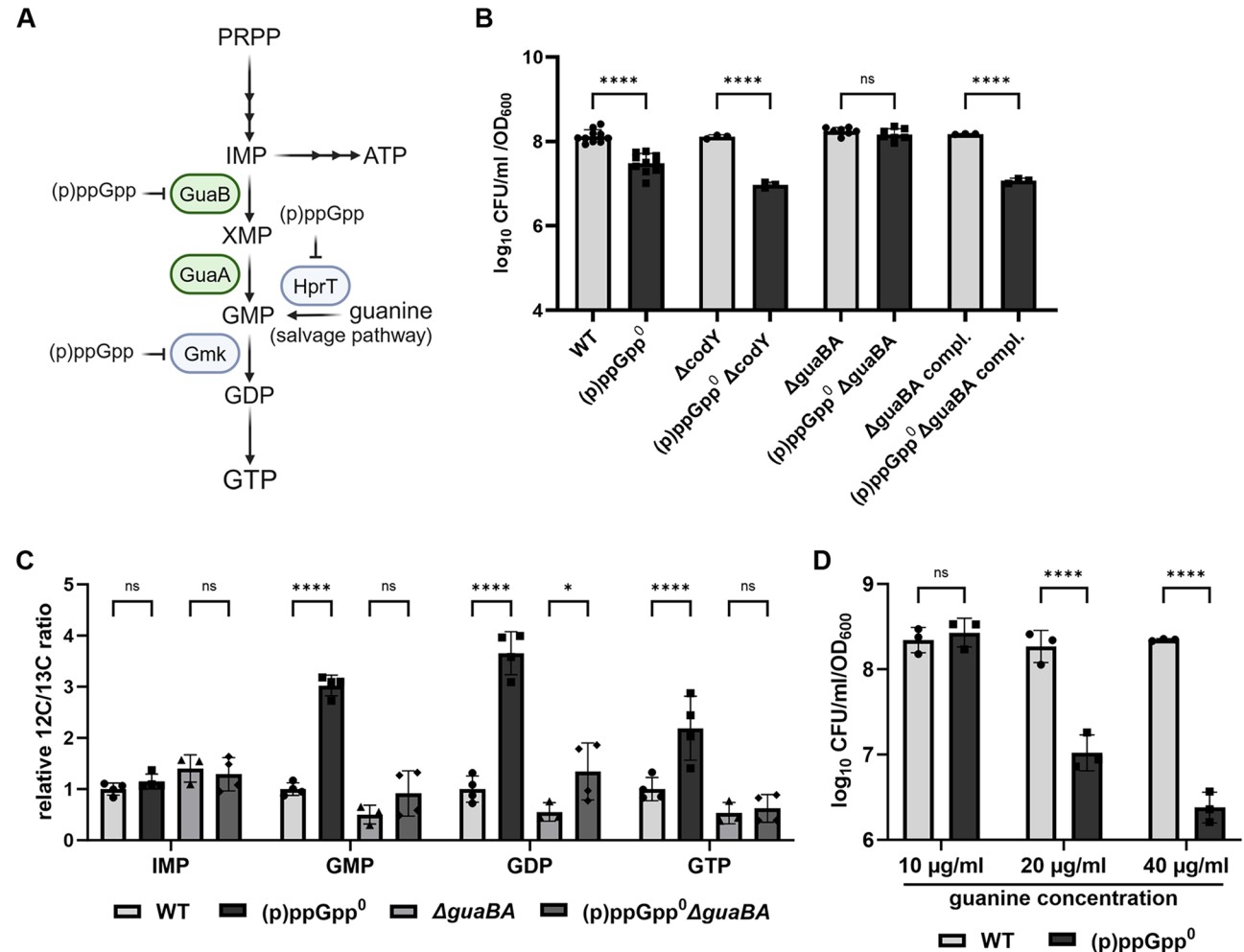

**Fig. 2 | (p)ppGpp is essential for GTP homeostasis during stationary phase starvation. A** The purine biosynthesis pathway is regulated by (p)ppGpp repressing the biosynthetic enzymes GuaB, HrpT and Gmk. **B** Strains were grown in CDM for 24 h and survival was determined by CFU/ml enumeration normalized to $OD_{600}$. Compl indicate that strains were complemented with the *gua* operon. Data shown are mean ± SD ($n \geq 3$). Statistical significance was determined by one-way analysis of variance (ANOVA) with a Tukey's post-test on $\log_{10}$ transformed data (****$p$-value < 0.0001, ns $p$-value = 0.9993). **C** Relative nucleotide levels were determined from late stationary phase cells by LC-MS/MS and normalized to $OD_{600}$. Data shown are mean ± SD ($n = 4$). Statistical significance was determined by two-way analysis of variance (ANOVA) with a Tukey's post-test (****$p$-value < 0.0001, *$p$-value = 0.0147, ns >0.1). **D** HG001 wildtype (WT) and (p)ppGpp[0] were grown in CDM with varying concentrations of guanine for 24 h and culturability was determined by CFU/ml enumeration normalized to $OD_{600}$. Data shown are mean ± SD ($n = 3$). Statistical significance was determined by two-way way analysis of variance (ANOVA) with a Šidák post-test (****$p$-value < 0.0001, ns $p$-value = 0.89).

bactericidal effect in the wild type. However, in the (p)ppGpp[0] mutant significantly less CFU were recovered upon mupirocin challenge. Thus, (p)ppGpp also protects from stress exerted by amino acid starvation in the exponential growth phase (Fig. S1B).

(p)ppGpp can also be provided by RelP or RelQ which are not activated via mupirocin. To ascertain which of the synthetases is primarily responsible for the stationary phase survival, we complemented the triple (p)ppGpp[0] mutant with the single enzymes. Expression of Rel or RelQ partially complemented the triple mutant indicating that both of these enzymes contribute to the (p)ppGpp-dependent survival advantage in the stationary phase (Fig. 1D).

In summary, these data show that (p)ppGpp is essential for bacterial culturability in the stationary phase as well as under induced stress conditions in exponentially growing bacteria.

### (p)ppGpp-dependent GTP homeostasis is essential for stationary phase culturability independent of CodY

It is well established that the synthesis of (p)ppGpp results in a reduction of the GTP pool[14,15] and that the activity of the GTP sensory protein CodY is severely altered upon (p)ppGpp synthesis[16]. To assess whether alteration of

the GTP pool contributes to the observed survival effect, we assessed the viability of different mutants in which GTP pathways were disturbed. The pleiotropic repressor CodY, when loaded with GTP, inhibits the expression of several biosynthesis-related gene clusters and virulence genes. We therefore hypothesized that the decreased survival of the (p)ppGpp[0] mutant might be ascribed to diminished expression of CodY target genes. However, the deletion of *codY* did not increase the culturability of the (p)ppGpp[0] mutant (Fig. 2B).

If the survival effect is indeed mediated via GTP any alteration of the GTP level should impact bacterial survival independent of (p)ppGpp. Indeed, deletion of the GTP synthesis operon *guaBA* rescued the survival defect of the (p)ppGpp[0] mutant. Chromosomal complementation with the *xpt-pbuX-guaB-guaA* operon reversed the protective effect imposed by the Δ*guaBA* mutation (Fig. 2B). We used LC-MS/MS analysis to confirm the anticipated nucleotide levels in the (p)ppGpp[0] and Δ*guaBA* mutants (Fig. 2A, Fig. S2A, Fig. S2A–C). Levels of ATP and the energy charge were decreased in (p)ppGpp[0] mutants independent of *guaBA* (Fig. S2). All guanine nucleotides (GMP, GDP and GTP) were significantly elevated in the (p)ppGpp[0] mutant (Fig. 2C, Supplementary Data S3), likely due to absence of (p)ppGpp-dependent inhibition of biosynthetic enzymes, including

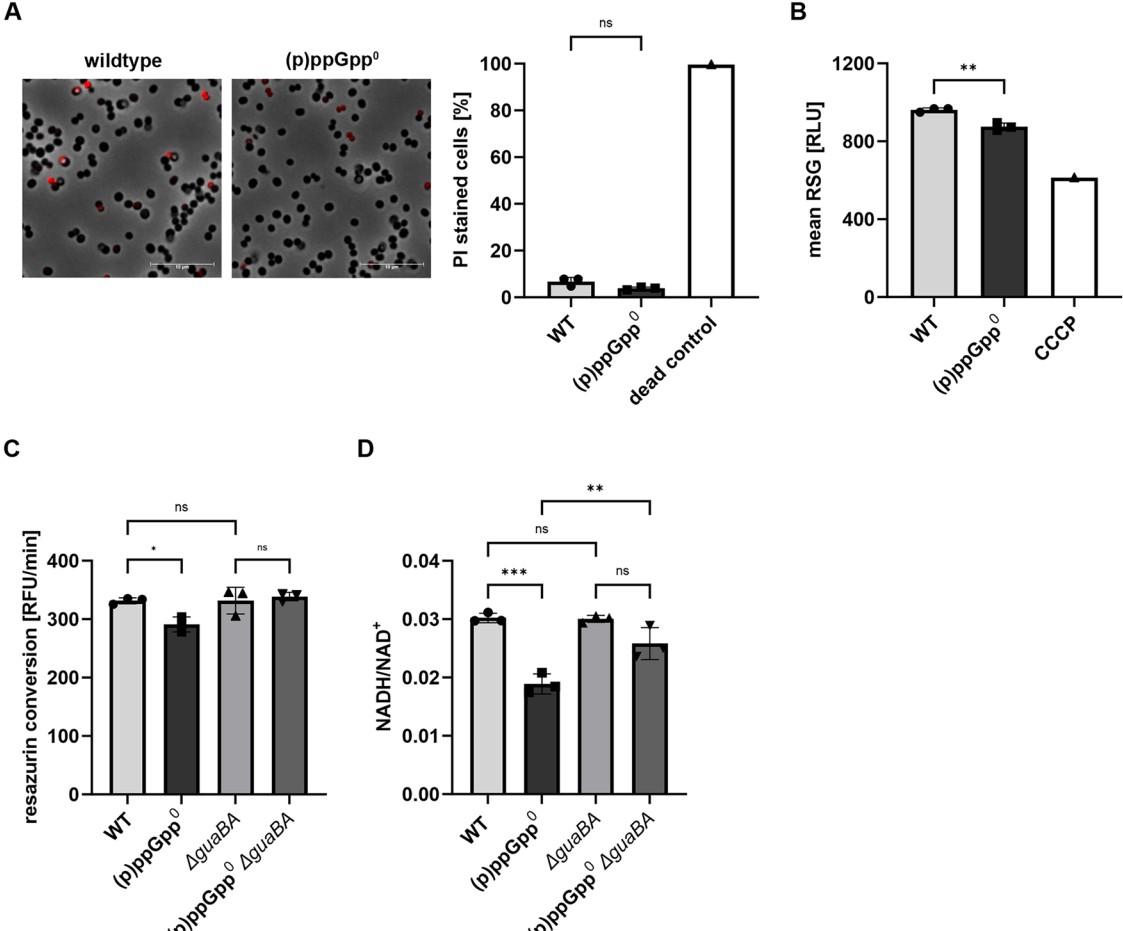

**Fig. 3 | The (p)ppGpp⁰ mutant maintains membrane integrity but displays decreased metabolic activity.** Strains were grown to late stationary phase (24 h) in CDM. **A** Representative fluorescence microscopy images of propidium iodide (PI)-stained HG001 wildtype (WT) and (p)ppGpp⁰ are shown. The percentage of PI-stained cells was measured by flow cytometry. For dead control bacteria were permeabilized with 70% EtOH. Data shown are mean ± SD (n = 3 biological replicates, n = 1 for dead control). Statistical significance was determined by one-way analysis of variance (ANOVA) with Tukey's post-test (ns p-value = 0.1022). Scale bar: 10 μm (**B**) Bacteria were stained with RedoxSensor™ Green reagent and analyzed by flow cytometry to monitor metabolic activity. Data shown are mean of RSG fluorescence ± SD (n = 3 biological replicates, n = 1 for CCCP control). Statistical significance was determined by one-way analysis of variance (ANOVA) with Tukey's post-test (**p-value = 0.0055). **C** Redox potential was accessed by resazurin conversion over time in HG001 wildtype, (p)ppGpp⁰, ΔguaBA and (p)ppGpp⁰ ΔguaBA. Data shown are mean ± SD (n = 3 biological replicates). Statistical significance was determined by one-way analysis of variance (ANOVA) with Tukey's post-test (*p-value = 0.029, ns p-value = 0.999 for guaBA versus WT, 0.9284 for pppGpp⁰/guaBA versus WT). **D** NADH/NAD⁺ ratio levels were determined using the NAD/NADH-Glo™ assay. Data shown are mean of ± SD (n = 3 biological replicates). Statistical significance was determined by one-way analysis of variance (ANOVA) with Tukey's post-test (***p-value =< 0.0002, **p-value = 0.0046, ns p-value = 0.999 for guaBA versus WT, 0.0603 for (p)ppGpp⁰ guaBA versus guaBA).

GuaB (Fig. 2A). This resulted in a prominent increase in the GTP/ATP ratio in (p)ppGpp⁰ mutant (Fig. S2). As anticipated, deletion of guaBA in (p)ppGpp⁰ resulted in a reduction of the guanine levels, reaching a level comparable to that observed in the wild-type strain (Fig. 2C). Thus, alteration of the GTP level and the GTP/ATP ratio was highly correlated with the survival phenotype of the strains.

We next manipulated levels of GTP via supplementation with guanine, which feeds into the salvage pathway of GTP synthesis (Fig. 2A). This resulted in a dose-dependent decrease in the culturability of the (p)ppGpp⁰ mutant (Fig. 2D).

The results confirm the hypothesis that elevated guanine nucleotide levels impede the culturability of stationary phase bacteria. However, this observation is not mediated by the GTP-dependent repressor CodY.

## (p)ppGpp-deficient cells enter a division-incompetent but viable state during starvation

To further investigate the cause and mechanism of the decreased culturability of the (p)ppGpp⁰ mutant, we used different viability assays based on

membrane integrity or metabolic activity. Despite the large decrease in CFUs during the stationary phase (Fig. 1B and S1A), only a small percentage of (p)ppGpp⁰ mutant cells were stained with the membrane-impermeable dye propidium iodide, and there was no significant difference between wildtype and the (p)ppGpp⁰ mutant (Fig. 3A and S7). Analysis of the metabolic activity of late stationary phase cells by RedoxSensor™ Green staining, which monitors bacterial reductase activity, revealed slightly reduced metabolic activity in the (p)ppGpp⁰ mutant (Fig. 3B). However, the decrease in metabolic activity was less than expected from the difference in CFU counts between wildtype and (p)ppGpp⁰ during the late stationary phase (only approximately 10% culturable (p)ppGpp⁰ mutant). Resazurin conversion (Fig. 3C), an indicator of cellular redox potential, was also decreased in the (p)ppGpp⁰ mutant compared to wildtype and the guaBA-negative strain. The measured NADH/NAD⁺ ratio was reduced in the (p)ppGpp⁰ mutants as well (Fig. 3D).

The discrepancy between the decreased CFU counts and low propidium iodide staining, together with slight decrease in metabolic activity in the late stationary phase, suggest that the (p)ppGpp⁰ cells did not immediately lose viability. It is more likely that the cells entered a dormant state,

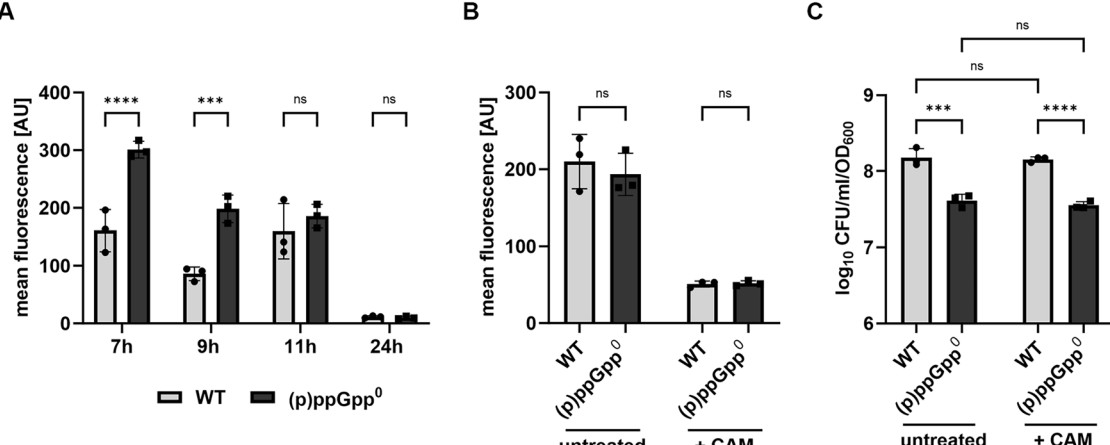

**Fig. 4 | Increased translation in (p)ppGpp⁰ mutant does not contribute to entrance into dormant state. A** Protein synthesis was monitored at different timepoints during growth in wildtype (WT) and (p)ppGpp⁰ cells by labeling with the puromycin analogue OPP and subsequent flow cytometry analysis. Data shown are mean of ± SD ($n$ = 3 biological replicates). Statistical significance was determined by two-way analysis of variance (ANOVA) with Šidák's post-test (****$p$-value < 0.0001, ***$p$-value = 0.0002, ns $p$-value = 0.623 for 1 h, 0.999 for 24 h). **B** Wildtype and (p)ppGpp⁰ cells grown to transition phase (7.5 h from inoculation time point)

were treated with 100x MIC chloramphenicol (+CAM), and protein synthesis by OPP incorporation and (**C**) culturability were assessed by CFU/ml/OD₆₀₀ enumeration. Data shown are mean of ± SD ($n$ = 3 biological replicates). Statistical significance was determined by two-way analysis of variance (ANOVA) with Šidák's post-test (on log₁₀ transformed data for (**C**) B: ns $p$-value = 0.6298 for untreated, 0.9987 for CAM, C: ***$p$-value = 0.0001, ****$p$-value < 0.0001, ns $p$-value = 0.9273 for WT untreated versus CAM, 0.9273 for pppGpp⁰ untreated versus CAM).

slowing their metabolic activity and halting cell division, resulting in the observed large decrease in CFU.

## The division-incompetent state of (p)ppGpp⁰ cells is not the result of increased translation during the stationary phase

(p)ppGpp is known to efficiently inhibit protein synthesis[9,23] and thereby may prevent the accumulation of protein aggregates[24]. We measured protein synthesis at different time points during growth by incorporating the puromycin analogue O-propargyl-puromycin (OPP) into newly synthesized proteins. As expected, in the (p)ppGpp⁰ mutant, translation was significantly higher in the late exponential and early stationary growth phases (7 h and 9 h) (Fig. 4A). However, in the late stationary growth phase (24 h), a complete shutdown of protein synthesis was observed, indicating that this occurred independent of (p)ppGpp. Delayed or absent translation inhibition can lead to the formation of protein aggregates[24]. However, only a slight increase in protein aggregates was observed in the (p)ppGpp⁰ mutant, as assessed by densitometric quantification of Coomassie-stained protein aggregates isolated from wildtype and (p)ppGpp⁰ late stationary phase cultures (Fig. S3A&B).

To further analyze whether increased translation in (p)ppGpp⁰ might contribute to entry into a division-incompetent state, we inhibited translation with 100x the MIC of the bacteriostatic antibiotic chloramphenicol during the transition to the stationary phase after 7.5 hours of growth. Chloramphenicol treatment almost completely inhibited protein synthesis in both wildtype and (p)ppGpp⁰ cells (Fig. 4B). However, chloramphenicol-induced inhibition of translation did not increase the culturability of (p)ppGpp⁰ cells (Fig. 4C).

Thus, translation inhibition appears to be delayed in the (p)ppGpp⁰ mutant, leading to slightly increased protein aggregation during the stationary phase. However, this delay in translation inhibition does not seem to be the cause for the nonculturable state of (p)ppGpp⁰ cells during stationary-phase starvation.

## Division-incompetent cell state of (p)ppGpp⁰ cells is oxygen dependent

In *S. aureus,* (p)ppGpp was shown to contribute to tolerance to oxidative stress[25], suggesting that elevated oxidative stress is a potential cause of the "dormant" state of (p)ppGpp⁰. To test this hypothesis, we analyzed

culturability under anaerobic and oxygen-limited, static (biofilm) conditions. Under these conditions, the culturability of the (p)ppGpp⁰ mutant was not compromised (Fig. 5A). However, treatment of aerobically grown cultures with the ROS scavenger N-acetyl cysteine (NAC) did not restore culturability of the (p)ppGpp⁰ mutant (Fig. 5B). No notable difference in endogenous ROS formation was detected between wildtype and the (p)ppGpp⁰ mutant during the stationary growth phase (Fig. 5C). Thus, (p)ppGpp-deficient cells were impaired in culturability only under aerobic conditions. However, we could not link this phenotype to ROS formation.

## Membrane function and architecture are impaired in (p)ppGpp⁰ cells

We next investigated whether aerobic respiration and linked proton motive force (PMF) directly impact bacterial survival. We measured membrane potential by staining with the carbocyanine dye DiOC₂(3). During the exponential growth phase, there was no difference in membrane potential between wildtype, (p)ppGpp⁰ and *guaBA*-negative cells (Fig. 6A). However, in the stationary phase, the membrane potential of the (p)ppGpp⁰ mutant was significantly lower than that of wildtype. In contrast, there was no significant difference in the *guaBA*-negative strain, indicating that high levels of GTP disturb the membrane potential. The PMF is composed of the electrical potential ΔΨ and the transmembrane proton gradient ΔpH. The intracellular pH was similar between wildtype and the (p)ppGpp⁰ mutant (Fig. 6B), indicating that it is the membrane potential ΔΨ which is mainly reduced in cells with high GTP levels.

Analysis of the membrane and cell wall architecture by co-staining with the membrane dye FM4-64X and the BODIPY-vancomycin conjugate revealed irregular staining of the (p)ppGpp⁰ cell membrane during starvation in the stationary phase, which was not observed in wildtype cells (Fig. 6C). In contrast, no abnormalities were detected in the cell wall of (p)ppGpp⁰ cells during stationary phase starvation compared to wildtype cells (Fig. 6C). The irregularities in the cell membrane of the (p)ppGpp⁰ strain, together with increased membrane fluidity, as measured by DPH fluorescence polarization (Fig. 6D), are indicative of membrane patches with increased fluidity.

Taken together, the results show that in the stationary phase, high levels of GTP, as found in the (p)ppGpp⁰ mutant, led to decreased membrane potential, changes in cell membrane architecture and increased membrane

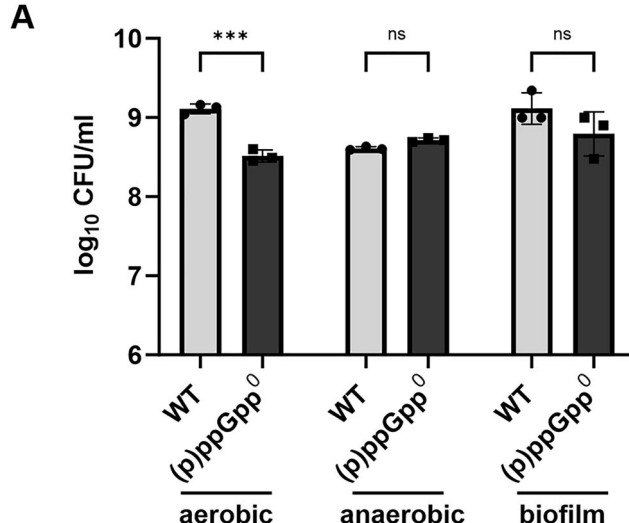

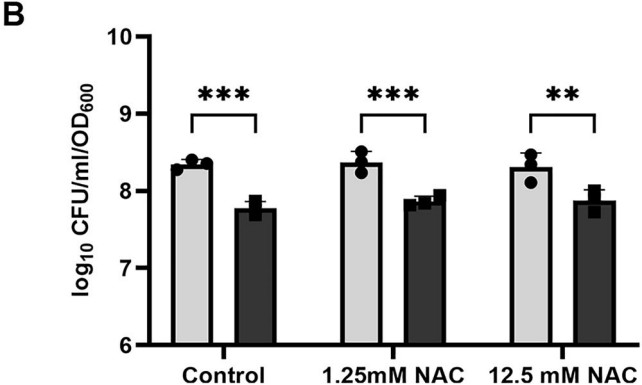

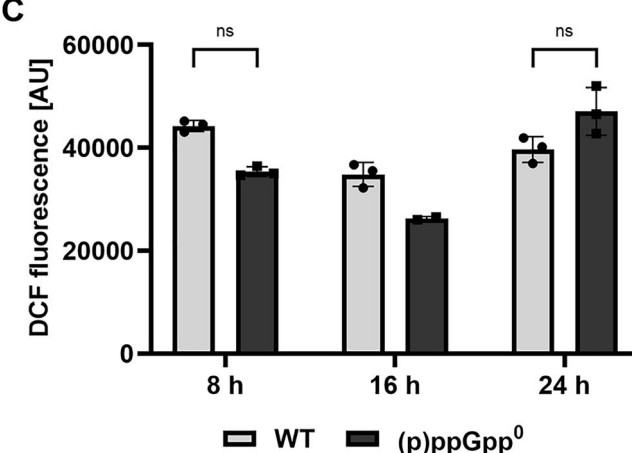

**Fig. 5 | Oxidative stress in (p)ppGpp⁰ mutant does not contribute to entrance into dormant state. A** HG001 wildtype (WT) and (p)ppGpp⁰ strain were grown under aerobic, anaerobic or oxygen-limited, static (biofilm) conditions for 24 h and culturability was accessed by CFU/ml enumeration. Data shown are mean ± SD ($n = 3$ biological replicates). Statistical significance was determined by two-way analysis of variance (ANOVA) with Šidák's post-test on $\log_{10}$ transformed data (***$p$-value = 0.0009, ns $p$-value = 0.7702 for anaerobic, 0.057 for biofilm). **B** Wildtype and (p)ppGpp⁰ cells were treated with N-acetylcysteine (+NAC) during transition to stationary phase (7.5 h) and CFU/ml counts normalized to $OD_{600}$ were determined before and after treatment. Data shown are mean of ± SD ($n = 3$ biological replicates). Statistical significance was determined by two-way analysis of variance (ANOVA) with Šidák's post-test (***$p$-value = 0.0003 for control, 0.0008 for 1.25 mM NAC, **$p$-value = 0.0029). **C** Monitoring of culturability, as determined by CFU/ml enumeration, and intracellular ROS levels, as measured by using the DCFH2-DA dye, which is oxidized to DCF by ROS, from early to late stationary phase. Data are shown as mean ± SD ($n = 3$ and $n = 2$ for 16 h). Statistical significance was determined by two-way analysis of variance (ANOVA) with Šidák's post-test, (ns $p$-value = 0.985 for 5 h, 0.9784 for 24 h).

fluidity. Thus, the decreased culturability of the (p)ppGpp⁰ mutant is likely due to loss of membrane potential.

## Impact of a lack of (p)ppGpp biosynthesis and GTP imbalance on the transcriptome during stationary phase starvation

To better understand how changes in GTP levels might alter membrane potential, we analyzed changes in gene expression in wildtype (GTP low), (p)ppGpp⁰ mutant (GTP high), ΔguaBA mutant (GTP low) and (p)ppGpp⁰/ΔguaBA double mutants (GTP low) after 24 h of growth (Supplementary Tables S1 und S2). Comparison of the (p)ppGpp⁰ mutant compared to wildtype revealed 115 genes which were significantly up-regulated and 115 which were down-regulated in the (p)ppGpp⁰ mutant (log2-fold change of ≥ 2 and ≤ -2, $p$ value ≤ 0.001) (Fig. 7A upper columns, Table S2). The pattern of gene expression is like that found in previous analyses following amino acid restriction[13] or transcriptional induction of (p)ppGpp synthetases (see Fig. S4 & S1C)[11]: Most CodY target genes were downregulated in the (p)ppGpp⁰ mutant e.g., the small RNA *rsaD* (-185.8-fold) and genes involved in amino acid synthesis and transport (*aur* (-65,7-fold), genes (*brnQ2* (-6.1-fold), *gltBD* (-14.8-fold and -12.2-fold), *dapABDL* (-12.5-fold for *dapA*, the *opp-3ABCDEF* operon (-20.1-fold for *opp-3B*) and *ilvD* (-7.2-fold)). As expected for the stringent phenotype genes coding for ribosomal proteins or for PSMs were upregulated. Furthermore, *groEL* (2.4-fold) and *groES* (4.9-fold), which are part of the CtsR/HrcA regulon, were found to be upregulated, indicative of proteotoxic stress[26]. These findings are consistent with the slightly elevated protein aggregates detected in the (p)ppGpp⁰ mutant during stationary phase starvation (Fig. S4F).

In a *guaBA* negative background in which the GTP level remained low in both the wildtype and the (p)ppGpp⁰ mutant the identified stringent response genes (see above) were hardly affected or even oppositely regulated (see comparison (p)ppGpp⁰ ΔguaBA versus ΔguaBA (Fig. 7A, middle column). Thus, (p)ppGpp-dependent gene regulation is most evident in bacteria with alterered GTP levels.

To further assess the unique GTP-dependent changes in gene expression we analyzed the effect of the *guaBA* mutation in a (p)ppGpp⁰ negative background (comparison of (p)ppGpp⁰ ΔguaBA versus (p)ppGpp⁰). The gene expression was highly similar to the expression pattern of the wildtype/(p)ppGpp⁰ comparison (compare upper and lower column Fig. 7A). These results clearly show that GTP is the main signal for the regulation of stringent response genes. We focused on genes which showed GTP dependent changes in gene expression independent of CodY (genes without CodY binding motif, see data S2). Profound GTP dependent gene regulation was observed for phage-encoded genes indicating induction of phages φ11, φ12 and φ13 during the stationary phase under GTP high conditions. These observations agree with previous studies that found the ΦSa3 phage to be downregulated under guanine-limited conditions[27].

Genes involved in electron transport were prominently altered: GTP resulted in downregulation of the *mnh* operon (table S2) which is proposed to encode a redox-energized complex involved in PMF maintenance (Bayer, McNamara et al. 2006). Expression of the *narGHIJ* operon (coding for nitrate reductase) and the *pstSCAB* operon (putative phosphate transporter) in contrast were significantly upregulated. Gene assigned to "respiration and electron transport" are illustrated in Fig. 7B: Genes coding for heme synthetase (*ctaA*), succinate dehydrogenase cytochrome B558 (*sdhCAB*), heme A IX farnesyltransferase *(ctaB)*, cytochrome c assembly protein (*hemX*), the ATPase complex *atpIBEFHAGDC* and cytochrome aa3 quinol oxidase (*qoxABCD*) were all downregulated under high GTP conditions.

**Fig. 6 | Membrane function and architecture is altered in the (p)ppGpp⁰ mutant. A** Membrane potential was measured during mid-exponential and late stationary phase by staining with the carbocyanine dye $DiOC_2(3)$. Data shown are mean of ± SD ($n = 3$ biological replicates, $n = 2$ for (p)ppGpp⁰ *guaBA* in late stationary). Statistical significance was determined by two-way analysis of variance (ANOVA) with Šidák's post-test (****$p$-value < 0.0001, ns $p$-value = 0.999 for (p)ppGpp⁰ versus WT and (p)ppGpp⁰ *guaBA* versus *guaBA*) (**B**) Intracellular pH was determined by pHRodo Green AM staining of latestationary phase bacteria. Data shown are mean ± SD ($n = 3$ biological replicates). Statistical significance was determined by a two-tailed, unpaired $t$ test (ns $p$-value = 0.069). **C** Representative microscopic images of FM4-64X (cell membrane) and BODIPY-vancomycin conjugate (cell wall) double-stained *S. aureus* from late stationary phase (24 h). Scale bar: 10 μm **D** Membrane fluidity was measured by DPH fluorescence polarization in late stationary phase (24 h). Data shown are mean ± SD ($n = 6$ biological replicates). Statistical significance was determined by a two-tailed, unpaired $t$ test (****$p$-value < 0.0001).

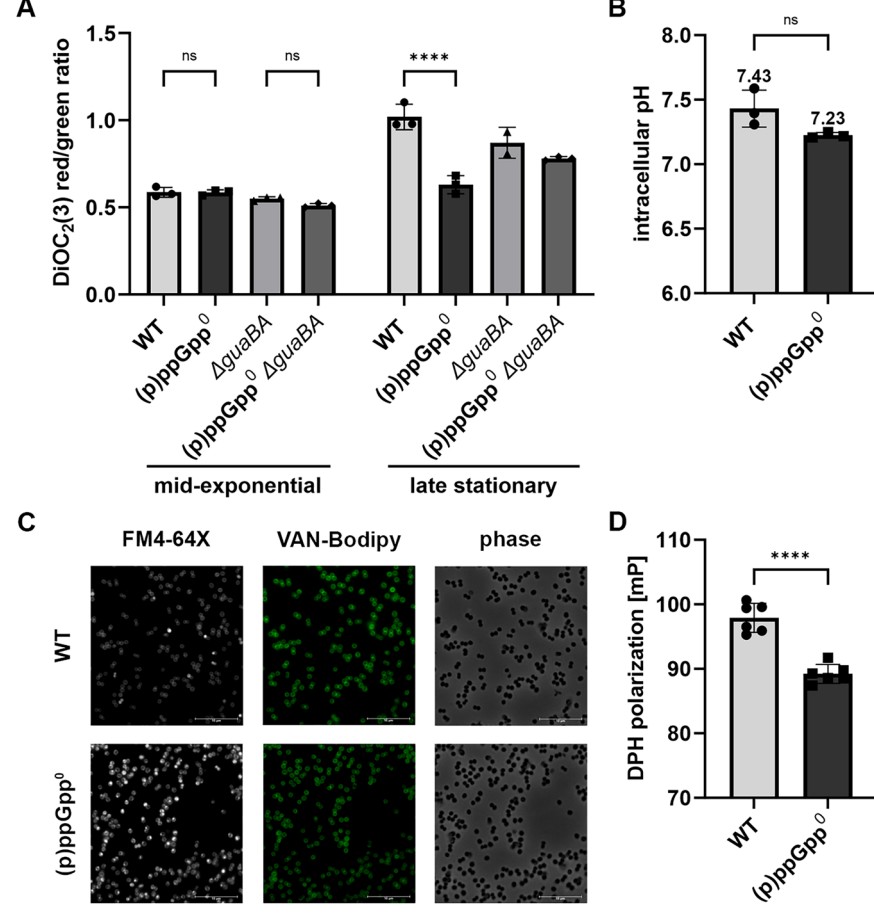

## The sensitivity of *qoxABCD* expression to GTP determines the activity of the electron transport chain and PMF

We assumed that the GTP-dependent down-regulation of the electron transport chain may account for the reduced membrane potential observed in the (p)ppGpp⁰ mutant (Fig. 6A). The terminal oxidase gene *qoxABCD*, a component of the electron transport chain (ETC) was prominently GTP regulated.

We verified the RNAseq data by RT-qPCR and confirmed decreased *qoxA* expression in (p)ppGpp⁰ but increased expression in the *guaBA*-negative strain compared to wildtype (Fig. 8B). Expression of sensitive genes can be controlled by the concentration of the transcription-initiating nucleotide (iNTP)[12]. Analysis of potential transcriptional start sites upstream of *qoxABCD* predicts an ATP as the transcription-initiating nucleotide (based on data from[28])(Fig. 7). Thus, we mutated the TSS + 1 to a GTP in wildtype and (p)ppGpp⁰ (named WT PqoxABCD_mut and (p)ppGpp⁰ PqoxABCD_mut) (Fig. 8A). This mutated TSS + 1 increased *qoxA* expression in the late stationary phase in (p)ppGpp⁰, indicating that this promoter is nucleotide sensitive (Fig. 8B). Alteration of the iATP to iGTP also resulted in greater CFU counts than those of (p)ppGpp⁰ (Fig. 8C). However, culturability was still lower compared to wildtype, showing that additional factors are likely to contribute to entry into dormancy.

To further demonstrate the importance of the PMF for stationary phase survival, we treated wildtype and (p)ppGpp⁰ cultures with sub-inhibitory concentrations of valinomycin during the late stationary phase and evaluated survival after 24 h. Valinomycin is an ionophore that disrupts membrane potential. Upon valinomycin treatment, wildtype and (p)ppGpp⁰ showed similar reductions in CFU counts compared to untreated cultures (Fig. 8D). Thus, disruption of membrane potential indeed compromises culturability and survival.

Together, these findings illustrate the significance of preserving PMF, particularly ΔΨ, throughout stationary phase starvation to sustain culturability.

## Preservation of PMF during starvation is crucial for antibiotic tolerance

We sought to determine whether the decreased membrane potential in the (p)ppGpp⁰ mutant would also result in a reduction of starvation-induced antibiotic tolerance. Therefore, we cultured wildtype and (p)ppGpp⁰ cells to exponential or stationary growth phase and exposed them to either 100x MIC oxacillin, ciprofloxacin or gentamycin. The minimal inhibitory concentration for these antibiotics was not different between wild type and (p)ppGpp⁰ and no significant difference in antibiotic survival was observed when bacteria were treated during exponential growth (Fig. 8E). Wildtype cells were barely killed by the antibiotics during the stationary phase, confirming the higher antibiotic tolerance of non-growing bacteria[3]. However, all antibiotics efficiently decreased the survival of the (p)ppGpp⁰ mutant during the stationary phase (Fig. 8F). Thus, under our growth conditions (p)ppGpp indeed contributes to antibiotic tolerance.

## Discussion

Here, we show that bacterial culturability in the stationary phase is dependent on (p)ppGpp-driven GTP depletion. Uncontrolled, high levels of GTP in the (p)ppGpp⁰ mutant resulted in membrane disorder and decreased PMF. The lower PMF could at least in part be linked to transcriptional downregulation of nucleotide-sensitive genes of the respiratory chain. Maintenance of respiratory chain activity under (p)ppGpp positive/low GTP conditions likely accounts for the observed tolerance of stationary phase cells to bactericidal antibiotics (Fig. 9).

**Fig. 7 | Transcriptomic changes in TCA cycle and respiration in dormant (p)ppGpp[0] cells.**
**A** Overview over all regulated genes with log2-fold > ± 2. "GTP-independent" compares the expression levels of (p)ppGpp[0] ΔguaBA vs. ΔguaBA, while "(p)ppGpp-independent" compares the expression levels of (p)ppGpp[0] vs. (p)ppGpp[0] ΔguaBA. The RNA-seq data shown are from three individual biological replicates (n = 3). Full RNA-seq data are available in Data S1 and S2 in the supplementary material. **B** Heatmaps displaying the RNA-seq fold change of selected genes involved electron transport chain and respiration (SEED annotation). Log₂ fold changes in relative transcript abundances are color-coded with red and blue, indicating up- and down-regulation, respectively. "GTP-independent" compares the expression levels of (p)ppGpp[0] ΔguaBA vs. ΔguaBA, while "(p)ppGpp-independent" compares the expression levels of (p)ppGpp[0] vs. (p)ppGpp[0] ΔguaBA. The RNA-seq data shown are from three individual biological replicates (n = 3). Full RNA-seq data are available in Data S1 and S2 in the supplementary material.

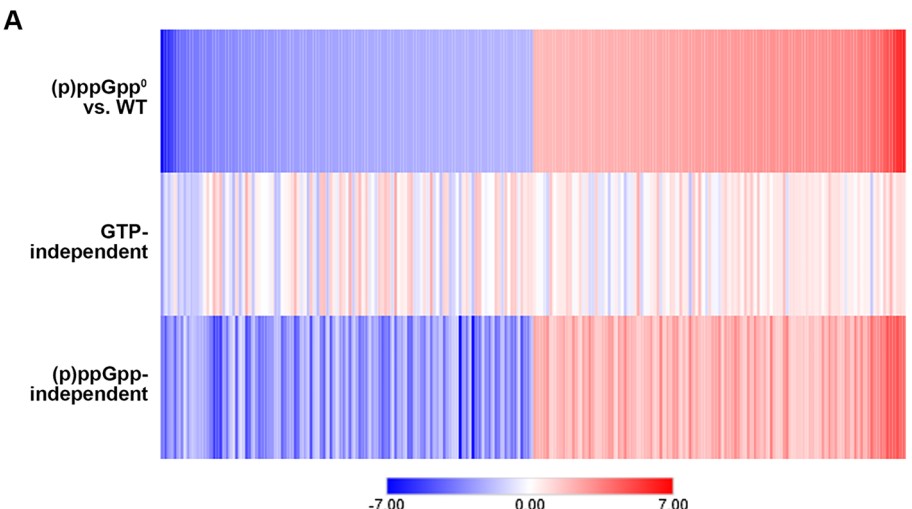

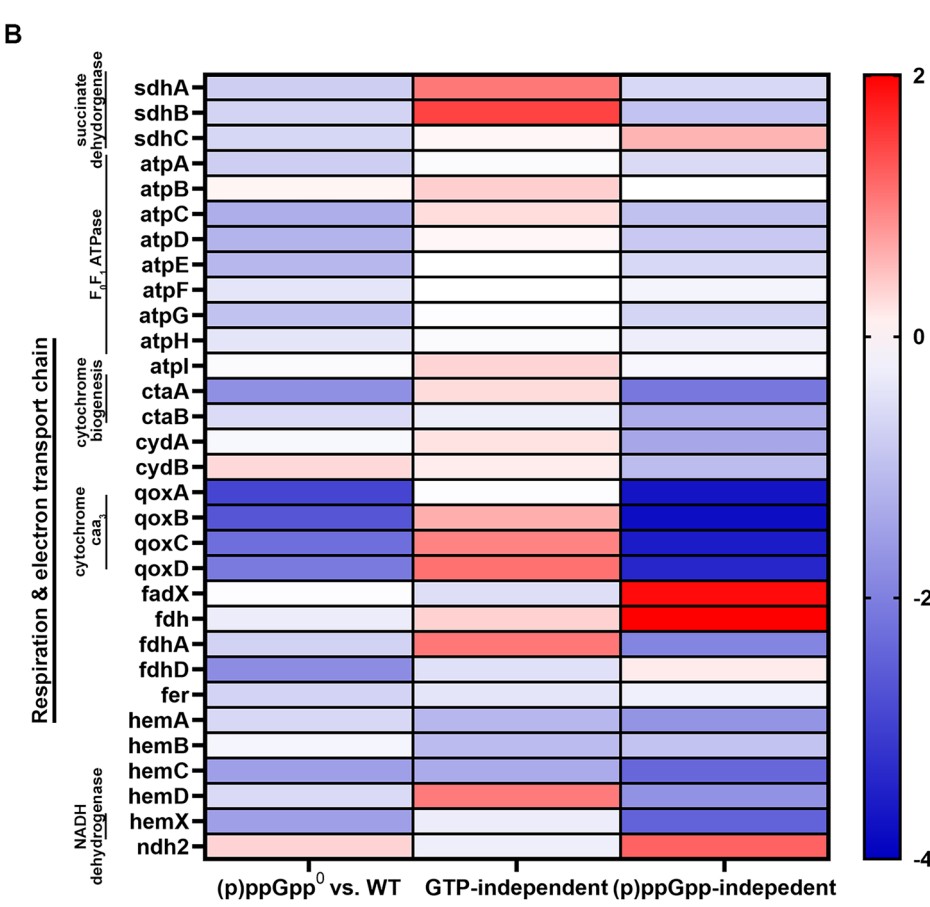

## (p)ppGpp exerts its effect via GTP homeostasis

The defect in culturability of the (p)ppGpp[0] mutant is clearly linked to an uncontrolled increase in GTP levels upon entry into the stationary phase. Culturability correlated inversely with changes in the GTP level induced by deletion of *guaBA* or by guanine feeding. Mutation of *guaBA* rescued the defect in the (p)ppGpp[0]. Thus, we conclude that the level of GTP homeostasis is the main determinant of bacterial culturability in stationary phase cells under our experimental conditions.

Control of guanine metabolism by (p)ppGpp is conserved in several Firmicutes species, such as *B. subtilis*,[14], *E. faecalis*[29] and *S. aureus*[15]. Recently, reduced long-term survival of the *S. aureus* USA300 (p)ppGpp[0] mutant[18] was linked to disrupted GTP homeostasis since suppressor mutants with

alterations in *gmk* were selected during long-term culture. In *B. subtilis*, low GTP levels reduce the growth rate but promote survival during amino acid starvation[14,30]. In this organism GTP accumulation in a (p)ppGpp[0] mutant results in a collapse of membrane potential and cell-death indicated by propidium iodide staining[31,32].

### In *S. aureus* perturbation of GTP homeostasis leads to induction of a VBNC state

Previously, it was proposed that (p)ppGpp depletion in *B. subtilis* results in "death-by-GTP" and cell death was thought to be due to unrestricted translation since (p)ppGpp results in severe inhibition of many proteins of the translational apparatus[9,14]. However, in *S. aureus*, uncontrolled

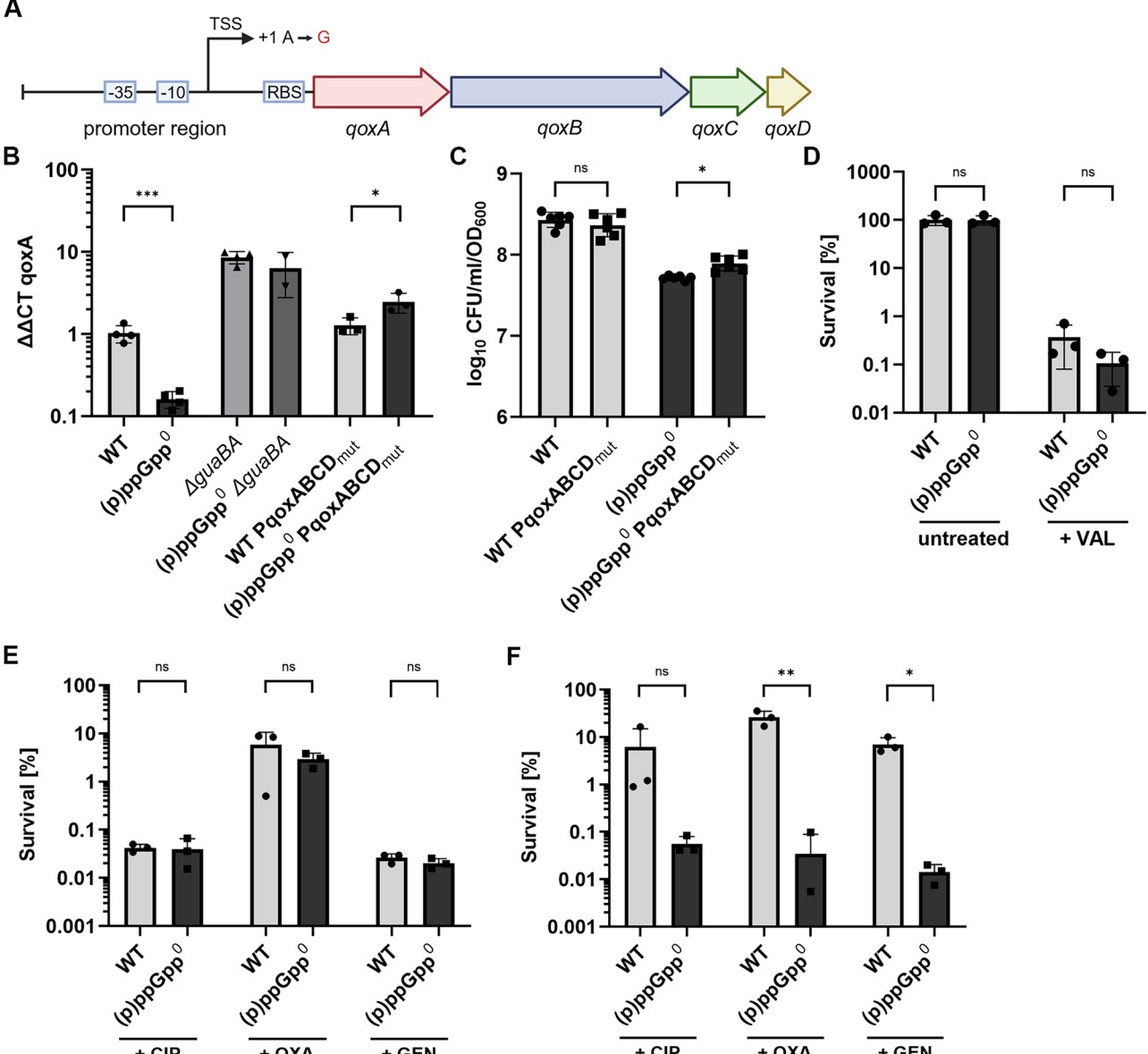

**Fig. 8 | Decrease in ETC activity contributes to entrance into dormant state and decreased antibiotic tolerance. A** *qoxABCD* gene operon with promoter region. The predicted initiation nucleotide (TSS + 1) iATP was mutated to iGTP. **B** Bacterial cells were harvested during the late stationary growth phase and total RNA was isolated. *qoxA* transcript levels were determined by RT-qPCR and normalized to *gyrB* expression by $\Delta\Delta Ct$ method. Data are shown as mean ± SD ($n = 3$ biological replicates, $n = 2$ for (p)ppGpp$^0$ *guaBA*). Statistical significance was determined by two-tailed unpaired *t* tests (\*\*\**p*-value = 0.0004, \**p*-value = 0.049). **C** Culturability during late stationary phase (24 h) was determined by CFU enumeration and normalized to OD$_{600}$. Data shown are mean of ± SD ($n = 6$ biological replicates from two individual experiments). Statistical significance was determined by a one-way analysis of variance (ANOVA) with Tukey's post-test (\**p*-value = 0.023, ns *p*-value = 0.657). **D** Wildtype (WT) and (p)ppGpp$^0$ cells were treated with

a sub-inhibitory concentration of valinomycin (20 µM) during the late stationary growth phase (24 h) and viability was assessed by CFU enumeration after 24 h of treatment. Survival was calculated in comparison to untreated samples. Data shown are mean of ± SD ($n = 3$). Statistical significance was determined by a two-way analysis of variance (ANOVA) with Šidák's post-test (ns *p*-value = 0.999 for untreated, 0.998 for VAL). **E** Mid-exponential phase or stationary phase (**F**) cells were treated with 100x MIC ciprofloxacin (+CIP), oxacillin (+OXA) or gentamycin (+GEN) and survival was calculated after 3 h (**E**) or 24 h (**F**) of treatment in comparison to untreated cells. Data shown are mean of ± SD ($n = 3$ biological replicates). Statistical significance was determined by a two-way analysis of variance (ANOVA) with Šidák's post-test (B: ns *p* value = 0.869 for CIP, 0.3404 for OXA, 0.2464 for GEN). C: ns *p*-value = 0.297 for CIP, \*\**p*-value = 0.008 for OXA, \**p*-value = 0.010 for GEN).

translation could not be linked to reduced bacterial survival. Inhibition of translation in the stationary phase occurred independently of (p)ppGpp. Furthermore, translation inhibitors did not restore the culturability of the (p)ppGpp$^0$ mutant. Our results also do not support the hypothesis that high GTP levels trigger a classical "cell death" programme. The (p)ppGpp$^0$ mutant of *S. aureus* was non-culturable but negative for propidium iodide uptake and maintained basal metabolic activity and transcription, albeit with significantly reduced ATP levels. This is indicative of the previously

defined state of VBNC (viable but non-culturable) or a deep dormant-like state[33]. It has been proposed that intracellular energy is the main factor maintaining the culturability of *E. coli* on solid media[24,34]. The (p)ppGpp mutant did indeed show a decrease in the energy charge which might contribute to VBNC formation. However, the (p)ppGpp$^0$ $\Delta$*guaBA* double mutant showed similar decrease in ATP and energy charge but was still culturable. We therefore hypothesize that the increased GTP level in (p)ppGpp$^0$ is the main cause of VBNC formation. Furthermore, it has

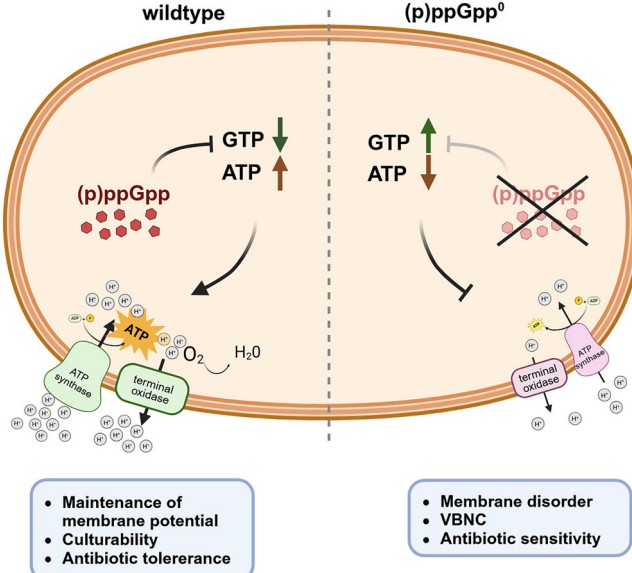

**Fig. 9 | (p)ppGpp-dependent regulation of GTP homeostasis enables maintenance of culturability.** Bacterial culturability in the stationary phase is dependent on (p)ppGpp-driven GTP depletion. Uncontrolled, high levels of GTP (low ATP) in the (p)ppGpp⁰ mutant resulted in membrane disorder and decreased PMF. The lower PMF could at least in part be linked to transcriptional downregulation of nucleotide-sensitive genes of the respiratory chain. Maintenance of respiratory chain activity under (p)ppGpp positive/low GTP conditions likely accounts for the observed tolerance of stationary phase cells to bactericidal antibiotics.

previously been shown that reducing only ATP in stationary phase *S. aureus* even increases growth after antibiotic challenge[3]. In *C. crescentus* fatty acid starved (p)ppGpp⁰ mutants also enter a VBNC-like state[35]. Lack of culturability was sought to be due to uncontrolled replication and thus the bacteria may be unable to recover from the miscoordination of cell cycle events. So far, we have no evidence for over-replication in the (p)ppGpp⁰ mutant. The transcriptional analysis even hinted to a down-regulation of some genes involved in DNA replication (e.g. *polC*) in the (p)ppGpp⁰ mutant. Thus, possibly the combined reduction of energy and high GTP levels (resulting in lower membrane potential) may severely impair the recovery after starvation. Our findings are consistent with previous observations in *B. subtilis*[14]. The results already indicated that GTP levels or the GTP/ATP ratio, but not ATP levels, correlate with the ability to withstand amino acid limitation.

## GTP-dependent lowering of the proton motive force induces VBNC state

We observed that the VBNC state observed in the (p)ppGpp⁰ mutant correlates with a decrease in the PMF. Inhibition of the PMF through valinomycin, a potassium ionophore which specifically dissipates the transmembrane potential mimicked the (p)ppGpp⁰ phenotype. Recently, the importance of PMF maintenance for stationary phase survival for *B. subtilis* was emphasized. It was shown that in the stationary phase the membrane potential was maintained at levels comparable to actively growing bacteria. Membrane depolarization by valinomycin resulted in severe inhibition of culturability[36]. It was proposed that depolarization of the membrane may disturb the normal localization of different peripheral membrane proteins. Furthermore, fatty acid starvation in a (p)ppGpp⁰ mutant of *B. subtilis* was shown to perturb the membrane potential and promote delocalization of the membrane protein MinD[32]. Consistent with these observations we also observed irregularities in the cell membrane of the (p)ppGpp⁰ strain. Similar membrane aberrations were observed in *S. aureus* treated with a membrane-damaging small molecule[37]. Whether these membrane changes and putative protein delocalization are caused by the

reduced PMF or by other GTP-dependent changes remains to be determined.

## (p)ppGpp is only required for survival under aerobic conditions

Under low oxygen conditions survival was (p)ppGpp independent and culturability was preserved in both the wildtype and the (p)ppGpp⁰ mutant. The survival under aerobic condition could be linked to the maintenance of the PMF for which the electron transport chain is crucial. An intact electron transport chain was recently shown to be important for the survival of aerobically grown *B. subtilis*. In this study, membrane depolarization was linked to ROS production and the Rieske factor QcrA. This iron-sulfur subunit of respiratory complex III seems to be a primary source of superoxide radicals[36]. *S. aureus* lacks Qcr and the terminal oxidase QoxABCD function as the main proton pump. Interestingly, the second messenger c-di-AMP is dispensable under anaerobic conditions but promote the survival under Qox-dependent respiration[38]. This could not be explained by protection from ROS alone. (p)ppGpp was shown to increase c-di-AMP synthesis possibly via the guanosine pathway[39,40]. Thus, the requirement for c-di-AMP and (p)ppGpp under aerobic growth may be functionally linked to proper Qox activity. This may help to maintain electron transport and allow the mutant to survive under aerobic conditions. Anaerobic respiration occurs independently of Qox via nitrate reductase NarGHIJ. Interestingly, *narGHIJ* expression is significantly upregulated in the (p)ppGpp⁰ mutant under high GTP conditions.

## (p)ppGpp/GTP dependent regulation via iNTP

Our RNAseq analysis revealed that genes coding for major components of the electron transport chain, including the terminal oxidase QoxABCD and CydAB as well as the $F_0F_1$ ATP synthase, were significantly downregulated under high GTP conditions. Accordingly, ATP levels (Fig. S2C) and membrane potential (Fig. 6A) were decreased in the (p)ppGpp⁰ mutant. We show that downregulation of the *qox*ABCD operon occurs through iNTP-dependent initiation of transcription. It was shown previously that some promoters appear to be sensitive to changes in initiating nucleoside triphosphate concentrations[12,15,41]. We mutated the predicted initiation of NTP transcription in the promoter region of *qoxABCD*, the most prominently downregulated component of the electron transport chain. By single point mutation (iATP to iGTP) of the native *qox* promoter we were able to restore wildtype levels of *qoxA* expression in the (p)ppGpp⁰ mutant (Fig. 8B). Furthermore, the culturability of the (p)ppGpp⁰ mutant was partially restored by this single nucleotide exchange. Expression of other genes involved in respiration and generation of membrane potential were also altered in the (p)ppGpp⁰ mutant. We assume that at least some of these genes/operons are also iNTP sensitive which may further contribute to the observed phenotype. We propose that (p)ppGpp (low GTP) in wildtype bacteria resulted in the preservation of the electron transport chain under aerobic conditions through iNTP-dependent promoter activity. Under high GTP/low ATP conditions these genes are repressed resulting in decreased PMF and thereby promoting the VBNC state.

## PMF maintenance during starvation is crucial for antibiotic tolerance

Here we showed that a (p)ppGpp⁰ mutant grown to the stationary phase lost antibiotic tolerance to the β-lactam oxacillin, the fluoroquinolone antibiotic ciprofloxacin and the aminoglycoside gentamycin. Non-growing wildtype bacteria are tolerant to antibiotics, as shown previously[3]. However, Conlon et al. did not find evidence that (p)ppGpp is involved in antibiotic tolerance[3]. The authors proposed that ATP depletion account for the observed tolerance of stationary phase cells. This seems to contrast our findings that (p)ppGpp-dependent maintenance of low GTP/high ATP levels protected the cells. This discrepancy might be explained by the experimental design. The growth medium used by Conlon et al.[3] likely does not induce a stringent response and/or might contain only very low amounts of guanine. Thus, under these conditions, the intracellular GTP pool might not be elevated in the (p)ppGpp⁰ mutant. The stationary phase tolerance is likely to be a

multifaceted phenomenon that may be overcome by different metabolic traits. The fine-tuning of nucleotide pools and/or nucleotide ratios seem to be crucial for antibiotic tolerance and resistance[40].

Our results emphasize the importance of PMF maintenance for stationary phase tolerance. Recent studies support that PMF is essential for starvation-induced tolerance in other organisms[42–44]. For P. aeruginosa it was proposed that active PMF supports antibiotic export (ampicillin, gentamycin) and thereby tolerance[5]. We found several transporters dysregulated in the (p)ppGpp[0] strain. The exact mechanism how GTP homeostasis and linked PMF maintenance contribute to antibiotic tolerance in S. aureus remains to be elucidated. High GTP and reduced membrane potential probably result in severe damage to the bacteria. Any additional stress could then lead to definitive cell death. Nevertheless, our findings support the idea of applying PMF inhibitors to combat antibiotic-tolerant bacteria including S. aureus[45,46].

## Materials and methods

### Growth conditions and CFU determination
The strains and plasmids used in this study are listed in the Supplementary Information, Table S1 and S2. S. aureus strains were grown in LB-Miller, tryptic soy broth (TSB) or chemically defined medium (CDM)[47] at 37 °C and 200 rpm. Unless otherwise indicated the CDM medium contained 10 µg/ml guanine.

For CFU experiments, bacteria from an overnight culture grown in LB-Miller were diluted to an initial optical density at 600 nm ($OD_{600}$) of 0.05 in fresh medium without antibiotics and grown with shaking (200 rpm) at 37 °C to the desired growth phase. LB-Miller medium was used for overnight culture since the (p)ppGpp[0] mutant in this medium showed no defect in survival. For oxygen-limited growth (biofilm conditions), bacteria were inoculated into 1 ml of fresh medium in a 24-well plate and statically grown at 37 °C, 24 h. For anaerobic growth conditions, 14 ml of fresh medium was inoculated into glass tubes, which were closed with a rubber stopper to minimize the oxygen supply and subsequently grown at 37 °C and 200 rpm. For certain CFU experiments, 400 µg/ml chloramphenicol (100x MIC of HG001 wildtype and (p)ppGpp[0]), 12.5 mM N-acetylcysteine or 20 µM valinomycin was added at the indicated timepoints during growth. For CFU/ml determination, a 10-fold dilution series was prepared in phosphate-buffered saline (PBS), 10 µl of each dilution was spotted onto TSB agar, and CFU counts were determined after 16-18 h of incubation at 37 °C. For microscopical determination of bacterial cell counts, bacteria were harvested by centrifugation, fixed in 3,7% formaldehyde for 15 min and bacterial cell counted using a Petroff-Hausser counting chamber. To adjust for slight differences in growth yield in different experimental settings the CFU/OD ratio are shown. For further information the CFU/ml for each experiment are shown in Fig. S6.

### Generation of S. aureus mutant strains
Oligonucleotides used for cloning, RT-PCR or verification are listed in Table S3.

### Generation of SH1000 (p)ppGpp[0]
The markerless relP single mutant, relQ single mutant, relP relQ double mutant, and the relP relQ rsh triple mutant [referred to as (p)ppGpp0] were obtained using the anhydrotetracycline-inducible suicide mutagenesis vector pKOR1. First, single mutants of relP and relQ were generated by deletions of regions containing known conserved domains responsible for the synthetic activities. Afterwards, the relQ mutation was introduced into the relP single mutant, resulting in the markerless relP relQ double mutant. Finally, the rsh mutation, with a deletion of the whole enzymatic N terminus, was introduced into the double mutant. For generating these mutants, plasmids pCG229 (Δ450–536 in relP), pCG230 (Δ343–429 in relQ), and pCG263 (Δ249–951 in rsh)[20] were transduced into SH1000. Plasmids were integrated into the chromosome at 43 °C overnight. One colony was picked, inoculated into 5 ml TSB and incubated at 30 °C overnight to facilitate plasmid excision. Cultures were then 10⁴-fold diluted spread on TSA

containing 2 µg/ml Anhydrotetracycline and incubated at 30 °C. All mutants were verified by sequencing amplicons spanning the mutation site.

### Transduction of guaBA complementation plasmid
The gua operon (xpt, pbuX, guaB, guaA) was amplified using oligonucleotides BamguaAc-for2 and BamguaAc-rev and cloned into the BamHI site of the integration vector pCL3. The plasmid was integrated into geh of strain CYl316 and transduced into S. aureus HG001 ΔguaBA and (p)ppGpp[0] ΔguaBA mutants.

### Mutagenesis strategy for WT-PqoxABCD_{mut} and (p)ppGpp[0]-PqoxABCD_{mut}
The plasmid pCG919mut was constructed for base substitution at the position +1 of the transcriptional start site of the $P_{qoxABCD}$ promoter. The complete promoter region with approximately 1000 bp of the left and right flanking regions was amplified in two PCRs using the primers pCG919gibfor and pCG919-2mutfor and pCG919-2mutrev and pCG919gibrev. The PCR products were subsequently cloned and inserted into the Bam HI-digested pIMAY-Z vector via Gibson cloning and transformed into E. coli IM08B. After verification of the base substitution (A → G) by sequencing, the plasmid was transformed into S. aureus HG001 wildtype and (p)ppGpp[0] cells by electroporation. The mutation was introduced into the genome by pIMAY mutagenesis[48,49] and verified by sequencing.

### RNA isolation, qRT–PCR and RNAseq
Bacteria were pelleted, resuspended in 1 ml of TRIzol (Thermo Fisher Scientific) and lysed using zirconia/silica beads (0.1 mm diameter) and a high-speed homogenizer[11]. RNA was isolated following the procedure recommended by the TRIzol manufacturer.

Relative quantification of psmα, rsaD, rpsL and qoxA transcripts by qRT-PCR was performed with QuantiFast SYBR Green RT-PCR Kit (Qiagen) using the Quantstudio3 system (Applied Biosystems). Briefly, 5 µg of total RNA was DNase-treated and diluted 1:10 for qRT-PCR. Relative expression of transcripts was calculated with the ΔΔCT method and normalized to gyrB gene expression.

For RNA-seq analysis, RNA isolated from the aqueous phase was further purified with an Amp Tech ExpressArt RNA Reading Kit. For each sample, a total of 100 ng of RNA was subjected to rRNA depletion, followed by cDNA library construction via IlluminaTM Stranded Total RNA Prep Ligation with a Ribo Zero Plus Kit according to the manufacturer's instructions. Libraries were sequenced as single reads (100 bp read length) using the NextSeq platform (Illumina) and NextSeq 500 Mid Output Kit v2.5. The sequences were demultiplexed with bcl2fastq (v2.19.0.316), quality checked with fastq (v0.20.1) and visualized with MultiQC (v1.7). Analysis of the RNAseq results was performed using CLC Genomic Workbench 23.0.1 (Qiagen). Reads were mapped to the reference genome of HG001 (NZ_CP018205.1). Differential gene expression was performed comparing (p)ppGpp[0] vs. wildtype, (p)ppGpp[0] ΔguaBA vs. ΔguaBA (GTP independent) and (p)ppGpp[0] vs. (p)ppGpp[0] ΔguaBA ((p)ppGpp independent). Genes with at least $\log_2$ fold change of >2 or <-2 and a p-value < 0.001 were defined as differentially regulated. Annotation of genes was performed according to the recent "Aureowiki" annotation of strain 8325 (http://aureowiki.med.unigreifswald.de/Main_Page)[50].

### Metabolite extraction and LC–MS/MS quantification
For metabolite extraction, 3 ml of bacterial culture was harvested by vacuum filtration, washed twice with 0.6% NaCl solution, transferred directly into 5 ml of quenching solution (acetonitrile:methanol:water, 40%:40%:20%, (v/v/v)) and shock frozen in liquid nitrogen. Afterwards, 1 ml of the cell suspension was lysed using zirconia/silica beads (0.1 mm diameter) and a high-speed homogenizer. The lysate was centrifuged at -9 °C at $20,000 \times g$ for 15 min, and the clear supernatant was stored at -80 °C for LC–MS/MS analysis. Relative concentrations of the nucleotides in the cell extracts were measured via isotope ratio LC–MS/MS.

The supernatant of metabolite extracts was mixed with $^{13}C$ internal standard in equal proportion and stored at −80 °C until analysis by LC-MS/M[51]. Samples were analyzed by LC-MS/MS, with an Agilent 6495 triple quadrupole mass spectrometer (Agilent Technologies)[52]. An Agilent 1290 Infinity II UHPLC system (Agilent Technologies) was used for liquid chromatography using a iHILIC-Fusion(P) (HILICON AB) column. The column oven was at 30 °C. LC solvents were: solvent A: water with ammonium carbonate (10 mM) and ammonium hydroxide (0.2%); solvent B: acetonitrile. The LC gradient was: 0 min 90% B, 1.3 min 40% B, 1.5 min 40% B, 1.7 min 90% B, 2 min 90% B. The flow rate was 0.4 mL/min. The injection volume was 3 μL. Settings of the ESI source were: 200 °C source gas, 14 L/min drying gas and 24 psi nebulizer pressure. The sheath gas temperature was at 300 °C and flow at 11 L/min. The electrospray nozzle was set to 500 V and capillary voltage to 2500 V.

For evaluation, the 12 C/13 C ratios were normalized to the $OD_{600}$ of the bacterial culture, and the relative abundances of each metabolite are shown.

## LIVE/DEAD and viability staining using microscopy and flow cytometry

Bacterial cultures were concentrated 10-fold and washed in 0.85% NaCl prior to staining for bacterial viability, as recommended by the manufacturer (Live/Dead BacLight™ Bacterial Viability Kit). All cultures were further diluted 5- to 10-fold to an $OD_{600}$ of 0.2-0.4 before staining with the Syto 9 and propidium iodide dye mixture (1:1 dye ratio). Fluorescence microscopy was performed with a Leica DM5500B fluorescence microscope using the following filter sets (excitation and emission filters): Syto 9: BP470 40-nm and BP525 50-nm; propidium iodide: BP535 50-nm and BP610 75-nm. Images were captured using a Leica DFC360FX monochrome camera with the following exposure times: 40 ms for Syto 9, 150 ms for propidium iodide and 6 ms for phase contrast channels. Images were viewed with in-built Leica ASF software. Alternatively, the percentage of propidium iodide-stained cells was analyzed via flow cytometry using a FACSCalibur flow cytometer (Becton Dickinson) (Fig. S7).

## RedoxSensor Green assay

To monitor bacterial reductase activity as an indicator of electron transport chain function, cells from the stationary phase were stained with Redox-Sensor™ Green reagent (BacLight™ RedoxSensor™ Green Vitality Kit) and analyzed by flow cytometry according to the manufacturer's instructions.

## AlamarBlue measurement

To measure cell viability via resazurin conversion, a stationary phase culture was tenfold diluted in PBS and mixed according to the manufacturer's instructions with alamarBlue™ reagent at a 1:10 ratio. After incubating at 37 °C, resofurin fluorescence (excitation: 540 nm, emission: 600 nm) was measured every 5 min. The increase in resofurin fluorescence over time was analysed via simple linear regression in GraphPad Prism 10.

## ATP and NADH/NAD$^+$ ratio determination

NADH/NAD$^+$ ratios were determined using the NAD/NADH-Glo™ assay (Promega) following the manufacturer's instructions. Luminescence was measured in a white 96-well plate using the Tecan Spark luminescence module.

## Click-it OPP protein synthesis labeling

Protein synthesis was quantified with a Click-it™ Plus OPP Alexa Fluor™ 488 protein synthesis labeling kit according to the manufacturer's instructions. Briefly, growing bacterial cells were incubated with 20 μM O-propargyl puromycin (OPP) for 30 min. Afterwards, the cells were harvested by centrifugation, fixed with formaldehyde and permeabilized with 70% ethanol (v:v) and lysostaphin. Then, OPP was conjugated to Alexa Fluor™ 488 using click chemistry according to the manufacturer's protocol. The stained cells were analysed using flow cytometry (Beckton Dickinson FACS Calibur).

## Detection and quantification of protein aggregation

Protein aggregates were isolated from 5 ml of culture. Bacterial cells were harvested by centrifugation and washed twice with PBS. Then, the cells were resuspended in buffer A (50 mM Tris, 150 mM NaCl, pH 8) and lysed using zirconia/silica beads (0.1 mm diameter) and a high-speed homogenizer at 6 m/s three times for 30 s each. The crude extract was centrifuged at $18,000 \times g$ for 30 min, after which the pellet was washed twice with buffer A containing 0.5% Triton X-100. The protein aggregates were then solubilized in rehydration buffer (7 M urea, 2 M thiourea, 4% (wt/vol) CHAPS, 100 mM DDT) and loaded on a 12% SDS–PAGE gel. Afterwards, the SDS–PAGE mixture was stained with InstantBlue™ Coomassie blue-based staining solution (Expedeon). Protein aggregates were densitometrically quantified from Coomassie blue-stained gels using the GelAnalyzer plugin of ImageJ software.

## ROS measurement

Endogenous ROS levels were measured using 2′,7′-dichlorodihydro-fluorescein diacetate (DCFH2-DA) dye based on a previous protocol[53]. DCFH2-DA is deacetylated by alkaline hydrolysis to generate DCFH2, which is oxidized by ROS to the fluorescent dye 2′,7′-dichlorofluorescein (DCF). Briefly, S. aureus HG001 wildtype and (p)ppGpp$^0$ cells were cultivated in CDM for the indicated durations and harvested at an $OD_{600}$ equivalent of $3 \times 10^8$ cells by centrifugation. The cell pellets were incubated with DCFH2 for 40 min. Relative DCF fluorescence was measured using a plate reader (Tecan Spark), with an excitation wavelength of 480 nm with a bandwidth of 20 nm and an emission wavelength of 530 nm with a bandwidth of 20 nm.

## Membrane potential measurement

Membrane potential was measured using a BacLight Membrane Potential Kit (Thermo Fisher). Cells were grown in CDM to the desired growth phase. The optical density was then adjusted to an $OD_{600}$ of 0.3 in PBS. $DiOC_2(3)$ was added to a final concentration of 30 μM, and the bacteria were incubated at 37 °C and gently agitated for 30 min. For plate reader measurements, 200 μl of stained cells was transferred to a Greiner Sensoplate (96-well, black, clear bottom). Using a Tecan Spark plate reader, red and green fluorescence was detected (excitation 488 nm, emission 535 nm and 600 nm), and the red:green fluorescence ratio was calculated. For flow cytometry, stained bacteria were pelleted by centrifugation, resuspended at $1 \times 10^7$ cells/ml in PBS and analysed using a BD FACSCalibur FL-1H and FL-3H.

## Intracellular pH determination

For intracellular pH determination, the fluorogenic probe pHRodo™ Green AM was used. S. aureus bacterial cells were grown to the stationary phase, and 0.6 ODU of the bacterial culture was harvested and washed with PBS. The cells were resuspended in 100 μl of pHRodo™ Green AM staining solution (final concentration of 10 μM) and incubated at 37 °C for 30 min. Afterwards, the cells were washed in 300 μl of PBS. Fluorescence was detected in a 96-well Sensoplate (Greiner) using a Tecan Spark plate reader (excitation: 500 nm, emission: 545 nm) or by flow cytometry using a BD FACS Calibur. For calibration of pHRodo Green AM, the "Intracellular pH Calibration Kit" was used according to the manufacturer's instructions.

## Cell membrane and cell wall co-staining

Formaldehyde-fixed S. aureus cells were stained with the fixable cell membrane dye FM4-64X (final concentration: 4 μg/ml) and the cell wall dye Bodipy™ Vancomycin-FL conjugate (final concentration: 1 μg/ml) for 10 min on ice. Fluorescence microscopy was performed using a Leica DM5500B fluorescence microscope and the following filter sets (excitation and emission filters): Bodipy™ Vancomycin-FL conjugate: 40-nm BP470 and 50-nm BP525; and FM4-64X: 50-nm BP535 and 75-nm BP610.

## Membrane fluidity measurement

Membrane fluidity was determined via DPH fluorescence polarization. Briefly, one OD unit of bacterial cells was harvested by centrifugation,

resuspended in prewarmed PBS containing 20 μM DPH (1,6-diphenyl-hexa-1,3,5-triene) and incubated at 37 °C for 15 min. Fluorescence polarization was measured with a BMG CLARIOStar plate reader in a 96-well Sensoplate (Greiner) (excitation: 360 nm-20 nm, emission: 450 nm-10 nm, dichroic filter: LP 410) at 37 °C.

## Antibiotic tolerance assay

To assess the antibiotic tolerance of *S. aureus* HG001 wildtype and (p)ppGpp$^0$ cells, either mid-exponential bacteria or late stationary phase bacteria were treated with 100 μg/ml oxacillin (100x MIC of HG001 wildtype and (p)ppGpp$^0$), 50 μg/ml ciprofloxacin (100x MIC of HG001 wildtype and (p)ppGpp$^0$), or 100 μg/ml gentamycin (100x MIC of HG001 wildtype and (p)ppGpp$^0$). Minimal inhibitory concentrations (MICs) were determined beforehand by the broth microdilution method or E-Test strips in accordance with EUCAST guidelines[54]. At the indicated time points, CFU counts were performed (as described under "CFU determination" and survival was calculated by comparison to $t_0$ ($t_{treated}/t_0$).

## Statistics and reproducibility

Data were analysed using GraphPad Prism 10. Relevant information regarding statistical tests was added to each figure caption where appropriate. For ANOVA, assumptions of homoscedasticity and normality of the residuals were checked visually by comparing the fitted values with the residuals and comparing the predicted values with the actual residuals. Post hoc tests were performed to conduct and correct for multiple comparisons. All tests for which this test was applied were two-tailed. A significance level cut-off of alpha = 0.05 was used. For visual purposes, asterisks denote significance levels as defined in figure captions. All data represent values from biological replicates (different cultures). The sample sizes are indicated in the figure captions. Statistical analysis was only performed when an experimental condition had at least $n = 3$ biological replicates.

## Reporting summary

Further information on research design is available in the Nature Portfolio Reporting Summary linked to this article.

## Data availability

All data are available from the GEO database under the accession number GSE254567. Source data are provided in Supplementary data 1-4.

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

## Acknowledgements
This work was funded by the Deutsche Forschungsgemeinschaft (Schwerpunktprogramm SPP1879 to CW (Project 423246275)) and by infrastructural funding from the Deutsche Forschungsgemeinschaft (DFG), Cluster of Excellence EXC 2124 'Controlling Microbes to Fight Infections' (Project 390838134) and Tübingen-Nottingham Seed core fund. Sequencing was performed by the Institute for Medical Microbiology (part of the NGS Competence Center NCCT (Tübingen, Germany), while data management, including data storage of the raw data for this project, was performed by the Quantitative Biology Center (QBiC), University of Tübingen, Germany. Mutants from the Nebraska library were obtained through the Network on Antimicrobial Resistance in Staphylococcus aureus (NARSA) program. Figures 2A, 8A and 9 were created with BioRender.com. We thank Libera Lo Presti for scientific discussions and editing the manuscript and Natalya Korn, Ellen Daiber and Vittoria Bisanzio for technical assistance. We acknowledge support from the Open Access Publication Fund of the University of Tübingen.

## Author contributions
Study design by A.S., C.W., Manuscript and figures by A.S. R.D., C.W., Transcriptomics by A.S., J.R. Metabolomic by J.R., H.L. Survival assays by A.S., S.I, L.S., P.I. Cloning by A.S, R.D. All other experiments by A.S.

## Funding

## Competing interests
The authors declare no competing interests.
