## [Transparent Peer Review file · Communications Biology]

(p)ppGpp-mediated GTP homeostasis ensures survival and antibiotic tolerance of *Staphylococcus aureus*

Corresponding Author: Professor Christiane Wolz

Version 0:

Reviewer comments:

Reviewer #1

(Remarks to the Author)

Please see the attached Word document.

Reviewer #2

(Remarks to the Author)

Salzer et al. investigated the role of (p)ppGpp-mediated GTP levels in *S. aureus* and bacterial survival at stationary phase. They studied the culturability, and tolerance of starved cells. Although this has been studied in a previous study (Reference 18), the authors have brought in extra experiments to delve deeper into the mechanistic insights by looking at the involvement of the transcriptional start site of cytochrome aa3 system (qoxABCD) and the proton motive force of the cells. With the work done in this manuscript, the authors claim that (p)ppGpp is essential for maintaining GTP homeostasis, preservation of electron transport and PMF for culturability and antibiotic tolerance.

Overall, the main findings of the paper are interesting, supported by data, and it will be of interest to many readers in the field. Further data in addressing below questions/comments would support the conclusions of this study further, and link the study into the context of literature to influence thinking in the field.

Specific comments:

- 1) Line 102-104, The authors hypothesise that deletion of purR in (p)ppGpp null mutant should lead to increased GTP levels, and this is why the culturability of this strain decreases further. Have the authors considered checking the GTP levels of this purR/(p)ppgpp null mutant strain? Including the GTP levels of this strain in the manuscript would strengthen their statement.
- 2) Line 107, and materials and methods – What is the concentrations of guanine used in this work? Was this Guanine hydrochloride or what version of this purine nucleotide was used in this study? Was the pH of the media kept the same when this nucleotide was added or did you need to re-adjust the pH by adding NaOH? If so, was there an alteration to the salt levels within the medium? This is lacking methodology details that are important for this to be reproducible by others.
- 3) Line 206, The threshold used for log₂ fold changes were >1 or <1, should the cut off be not more stringent (i.e. above 1.5 or even 2?) I recommend >2 and <2 to be used instead of >1, <1.
- 4) To further validate the RNA-Seq results, the authors should consider using some of the TCA-cycle mutants they refer to in their manuscript. They should be easily transducible into the (p)ppGpp null-background if their regulation is a cause for the GTP-dependent culturability. They claim that the downregulation of TCA cycle and respiration may account for the reduced membrane potential observed during stationary phase starvation. It would be ideal to prove this claim with further experiments utilising some of the TCA-cycle mutants. Additionally, their transcriptional start sites could also be analysed similar to that of qoxABCD studied in the manuscript.
- 5) Figure S5, What is the OD-CFU ratio of WT, (p)ppGpp-null and additional mutants used in this study? If you are inoculating the same starting OD for the tolerance assays, does that translate to the same CFU starting?

6) Figure 9, This model is not fully supported by data. There are arrows to TCA cycle, however authors have not proven this. Also, directly linking (p)ppGpp to antibiotic tolerance is too far of a stretch if only ciprofloxacin and oxacillin is tested. Please refine the model appropriately, or provide more work or references to support the arrows indicated.

7) There have been links of c-di-AMP and ppGpp crosstalk in the literature. *S. aureus* c-di-AMP mutant survival has been linked to the *qoxABCD* system in a suppressor analysis, potassium transport, aerobic respiration and membrane potential. Authors do not discuss any of such literature in the context of their new findings, which could add value to the discussion of this manuscript and influence some thinking in *Staphylococcus*, and nucleotide signalling research fields.

Additional comments:

8) Line 48 -> Typo - HrpT should be HprT.

9) Line 82, Why 17h and 24 h were chosen? How do you define mid-stationary? Have the authors looked at the culturability of the strains over a longer term than 24 hours?

10) Line 113, ...levels "of what?" were observed.....

Reviewer #3

(Remarks to the Author)

In this manuscript, "(p)ppGpp-mediated GTP homeostasis ensures the survival and antibiotic tolerance of *Staphylococcus aureus* during starvation by preserving the protonmotive force", by Professor Wolz and colleagues showed that (p)ppGpp-dependent restriction of GTP pool contributes to the culturability of starved cells and antibiotic tolerance. The manuscript demonstrates a commendable level of clarity in its presentation. The key findings are very well supported by experiments. The presentation of the data and figures exhibits clarity. However, some areas need attention or clarification.

1) Lines 45-49 are difficult to follow, mainly when the concept of (p)ppGpp synthesis correlates inversely with the GTP pool is mentioned. It takes multiple readings to connect this information with the previous lines.

2) In lines 75-76, please highlight the significance of the strains in the context of the study.

3) While lines 77-80, describe an experiment that was performed to induce the expression of Rel-dependent (p)ppGpp synthesis in (p)ppGpp^o mutant, line 69 reports that mutants were unable to synthesize (p)ppGpp. I have difficulty understanding the results of (p)ppGpp^o mutant. Also, supplementary S1B should have included the control group for both WT and mutant that has not been treated with mupirocin.

4) I request authors to explain the results Figure S1C particularly focusing on why a significant change is observed at 7 hours but not at 24 hours.

5) What could have been the possible reason for relP complementation to not show the effect?

6) Please mention the mutants in line 94, when they are first introduced.

7) Line 102-104, the authors concluded that deletion of *purR* resulted in elevated GTP levels that were detrimental to bacterial susceptibility. The authors did not measure the levels of GTP in this strain. How do they conclude that it is because of GTP and not ATP?

8) In Figure 2C, why no significant change was observed in WT vs (p)ppGpp^o mutant at 10µg/ml guanine concentration?

9) Please mention about CCCP control and its role in line 127.

10) I would appreciate it if author could explain why most of the experiments did not involve the *relSyn*⁻ or *relQ*-complemented strains.

11) Please include the level of statistical significance between Δ *guaBA* and (p)ppGpp^o Δ *guaBA* in figure 3E and between WT and (p)ppGpp^o in figure 8D.

12) Please mention the time point at which the experiment in figure 4b and 4c was conducted.

13) The significance of the current study is not adequately elucidated in the manuscript.

Version 1:

Reviewer comments:

Reviewer #1

(Remarks to the Author)

The organization of the response letter is scrambled and some internal communications were left in the document, which made it a little tricky to navigate. As for the manuscript, the authors performed some additional experiments to address our prior concerns. Given the personnel shortage in the lab, this is adequate. There are a few typographical errors in the current manuscript, some of these are listed below.

Line 81: please change (p)pppGpp to (p)ppGpp.

Line 327: please add the bracket "(" before (p)ppGpp.

Line 331: please add a hyphen between "VBNC" and "like."

Line 353: please italicize "*S. aureus*."

Line 430: how long were the biofilms cultured?

Reviewer #2

(Remarks to the Author)

The authors have addressed all my comments/concerns. The increased threshold of fold changes with additional experiments have tightened the findings and focus of this study. I have no further suggestions.

Reviewer #3

(Remarks to the Author)

I would like to thank the authors for the resubmission of the manuscript titled '(p)ppGpp-mediated GTP homeostasis ensures the survival and antibiotic tolerance of Staphylococcus aureus during starvation by preserving the proton motive force.' The authors have made significant effort to address the comments and suggestions provided by the reviewers. There are notable improvements in both the clarity and overall quality of the manuscript. I am satisfied with the changes made in the revision, though I still have few minor comments.

1. Please introduce (p)ppGpp in line 68 instead of line 70.
2. In line 81, authors mentioned general phenomena, I believe they intended to convey it is not a strain-specific phenomenon. Please revise the sentence.
3. In line 118, Fig 3E is mentioned, but there is no Fig 3E in the panel.
4. In line 121 and 125, Fig 2D is referenced; I believe the authors intended to refer to Fig 2C instead. Similarly, in line 129, fig 2C should be corrected to Fig 2D.
5. The scale used in Fig 1B and 1E differs in representation. Please correct for uniformity across figures.
6. There are several typographical errors in the manuscript, like line 345-PMN, line 327-(p)ppGPP, line 353-italicized S. aureus

Reviewer #4

(Remarks to the Author)

Answer to the Editor:

All the reviewers raised a couple of common concerns. The first concern is assessing the levels of GTP and its intermediates in the pur deletion background to provide experimental support for lines 102-103. Secondly, the claim that, “downregulation of TCA cycle and respiration may account for the reduced membrane potential observed during stationary phase starvation”, is based on transcriptome data. Hence, testing a few of the TCA cycle genes and assessing whether modifying their transcriptional start sites, like the quoxABCD experiments, will provide better clarity into how the lowering of PMF is a multifactorial event.

***Answer:** Dear editor, thanks a lot for the very careful considerations of our manuscript. Based on your and the very helpful reviewer’s comment we have now included several new data and have rewritten large part of the discussion. We deleted the PurR finding and the statements concerning TCA cycle activity. The PurR results are not really required for the present manuscript and will be subject to further investigation. Upon careful reanalysis of our data, we are now much less convinced that the TCA cycle activity plays a major role for survival.*

Other major concerns from reviewers are listed below and should be addressed:

Reviewer One:

1. Bacterial cells, depending upon their growth stage, undergo reductive cell division where DNA replication is initiated in cells under stress (like in stationary phase), however, these cells fail to divide, leaving them with multiple copies of DNA (PMID: 26798489 Titel anhand dieser Pubmed-ID in Citavi-Projekt übernehmen). Hence, the CFU or cell number will not increase as a result of reductive division depending upon the growth phase in which these cells are (PMID: 26798489 Titel anhand dieser Pubmed-ID in Citavi-Projekt übernehmen). As suggested, the authors may look into the cell number in each phase of growth for both WT and ppGpp0 mutant using fluorescent beads and plot their graphs in terms of CFU/mL

***Answer:** Thanks for these thoughts and the link to this interesting article which nicely describes the main features of the stationary phase of *E. coli*. Some of the described features may also apply to *S. aureus*. In our main experiments we analyzed bacteria after 24h of growth which is likely not yet the deep stationary phase. The onset of the stationary plateau is clearly visible through OD measurements (e.g. see Fig1A). To ascertain that the OD measurement indeed reflects bacterial numbers we performed additional microscopic analyses. In our growth conditions the OD value is highly concordant with the bacterial number enumerated by cell counting in the microscope. We now included the result of stationary phase wild type and (p)ppGpp mutant in the **new Fig. 1C**.*

*However, we prefer to show CFU/OD in the main figures. Under some conditions we are comparing strains/conditions with differences in the final OD. We feel that showing CFU/OD is more appropriate to show the “non-culturability”. Nevertheless, we added now the CFU/ml values for the experiments as **new Suppl. Fig 6**. The overall outcome of the experiment remains the same.*

*Concerning DNA content in stationary phase (chromosome copy numbers) we have no indication that there is a difference between wild type and (p)ppGpp mutant strain. We discuss now this point: “In *C. crescentus* fatty acid starved (p)ppGpp0 mutants also enter a VBNC like state 37. Lack of culturability was sought to be due to uncontrolled replication and thus the bacteria may be unable to recover from the miscoordination of cell cycle events. So far, we have no indication for over-replication in the (p)ppGpp0*

mutant. The transcriptional analysis even hinted to a down-regulation of some genes involved in DNA replication (e.g. polC) in the pppGpp0 mutant”.

2. The authors should include a *guaBA* complement strain in their physiological assays.

Answer: *We now complemented the *guaBA* mutation in wild type and pppGpp mutant. The complementation restored the *guaBA* phenotype, see new Fig. 2B*

3. The authors have talked about the VBNC nature of their ppGpp0 mutant in their discussion. However, they may consider adding a line or two on adenylate energy charge and its potential role behind their observation.

Answer: *We now added new results concerning the energy charge (new Fig. S2). The energy charge was only slightly decreased in the mutant strains. We have rewritten the part of the discussion accordingly:*

*“It has been proposed that intracellular energy is the main factor maintaining the culturability of *E. coli* on solid media 26,36. The (p)ppGpp mutant did indeed show a decrease in the energy charge which might contribute to VBNC formation. However, the (p)ppGpp0 Δ *guaBA* double mutant showed similar decrease in ATP and energy charge but was still culturable. We therefore think that the increased GTP level in (p)ppGpp0 is the major cause for VBNC formation. Furthermore, it has previously been shown that reducing only ATP in stationary phase *S. aureus* increases growth after antibiotic challenge 3. In *C. crescentus* fatty acid starved (p)ppGpp0 mutants also enter a VBNC like state 37. Lack of culturability was thought to be due to uncontrolled replication and thus the bacteria may be unable to recover from the miscoordination of cell cycle events. So far, we have no evidence for over-replication in the (p)ppGpp0 mutant. The transcriptional analysis even hinted to a down-regulation of some genes involved in DNA replication (e.g. *polC*) in the pppGpp0 mutant. Thus, possibly the combined reduction of energy and high GTP levels (resulting in lower membrane potential) may severely impair recovery after starvation. Our findings are consistent with previous observations in *B. subtilis* 14. The results already indicated that GTP levels or the GTP/ATP ratio, but not ATP levels, correlate with the ability to withstand amino acid limitation*

4. For the LC/MS data, the authors should provide the raw reads to better appreciate their results. They may consider citing proper references in support of the methodology that they have used for the same.

Answer: *We now added the raw data as additional data file S3 and described the method in more detail. can ask Johanna for a more detailed description of methodology*

5. The authors have pointed out that unrestricted translation is not responsible for the loss of culturability in the late stationary phase and linked this phenotype to differentially transcribed genes in the late stationary phase through transcriptome profiling. To investigate the transcript levels, the authors may consider looking at the transcript levels of ppGpp-influenced genes at different time points between WT and the mutant. As transcriptome analysis of the same may take time, the authors can use observation made in figure S1C for a wide range of genes.

Answer: *The first author already left the lab quite a while ago and the RNA is no more available for further analyses. We so far had no personal capacity to gain additional, publishable qRT-PCR data. However, a master student did already analyze gene expression of additional genes in the past and the results are general consistent with the RNAseq data. However, these data are not publishable in the present form (e.g. missing of independent replicates). Thus, we did not add these results in the present manuscript.*

6. The authors may like to provide evidence addressing the link between GTP levels and PMF perturbations (imbalance and restoration) wherever they have inferred so.

***Answer:** We tried to make this clearer. We have extensively rewritten the result and discussion*

7. The authors may like to provide experimental evidence on whether the PMF contributes to tolerance via efflux pumps or uptake of antibiotics like aminoglycoside (known to exhibit PMF-dependent uptake). [This concern was also raised by reviewer two]

Answer:** We now included additional data using the aminoglycoside antibiotic gentamycin. We observed similar to oxacillin and ciprofloxacin that the pppGpp mutant is significantly less tolerant. These data are now included in the **new Fig. 8

8. The rest of the comments are addressable as such and should be addressed.

Reviewer two: Comments from reviewer two complement reviewer one's comments. Overall, other comments by reviewer two will help improve the methodologies and results discussed in this manuscript and should be addressed. The authors should redraw their final model in Figure 9 for better elucidation (as per suggestions from reviewer two on TCA cycle genes).

Reviewer three: The comments from reviewer three overlap with those of the other two reviewers as well. Otherwise, the other concerns of reviewer three should be addressed as such. Additionally, the authors should consider elaborating on the significance of their current study in their discussion section (as pointed out by reviewer three).

Answer: For detail answers to reviewers see below

Answer to the reviewers

Reviewer 1

The stringent response, mediated by the alarmone (p)ppGpp, is well-conserved in bacteria and facilitates survival under amino acid starvation. In the opportunistic pathogen *Staphylococcus aureus* (p)ppGpp production has been proposed to maintain GTP homeostasis during starvation. In this study, the authors propose that (p)ppGpp production ensures that electron transport chain and proton motive force generation are sustained in nutrient-starved, late-stationary phase cultures. Loss of the stringent response reduces *S. aureus* culturability after 24 h of growth in chemically-defined media. In their ppGpp-null mutant, the authors show that guanine nucleotides are elevated, whereas NADH/NAD⁺ and ATP levels decrease. They further demonstrate late- stationary phase ppGpp-null mutants exhibit reduced tolerance toward a topoisomerase-targeting fluoroquinolone and a cell wall synthesis inhibitor.

Overall, the findings in this paper offer some new insight into how ppGpp is linked to culturability and antibiotic tolerance in nutrient-starved *S. aureus*, but it would benefit from revision. As stated below, some of the experimental approaches can be more rigorous to ensure that the findings that the authors report are robust and reproducible. From their transcriptomic analysis, the authors focus on only one operon, *qoxABCD*, which very marginally impacts the culturability of the ppGpp- null mutant. Exactly how the expression level of the *qox* operon, which is linked to aerobic respiration and is a key component of their model in Fig. 9, impacts the metabolism and respiration of the ppGpp-deficient strains is not thoroughly investigated.

Key Comments:

In a number of figures in which the authors are assessing colony forming units (CFUs) in the cultures, the authors normalize CFU/mL by OD600. For instance, this was done in Fig. 1C and 1D, but not 1B or 4C. The authors state that this normalization was done to account for culturability. However, this may not be the most accurate approach. Bacteria in stationary phase can undergo reductive division and are smaller than exponentially growing cells. The number of cells that are accounted for by dividing CFU/mL by OD600 may vary depending on the growth phase of the culture. It would be more accurate for the authors to account for the total number of cells in the culture using fluorescent counting beads. Given that Fig. 3A suggests that the ppGpp0 cells are not dead/dying, the authors should report CFU/mL for their figures instead of the OD-normalized ones.

Answer: *The onset of the stationary plateau is clearly visible through OD measurements (e.g. see Fig1A). To ascertain that the OD measurement indeed reflects bacterial numbers we performed additional microscopic analyses. In our growth conditions the OD value is highly concordant with the bacterial number enumerated by cell counting in the microscope. We now included the result of stationary phase wild type and (p)ppGpp mutant in the new Fig. 1C.*

We added now the CFU/ml values as suggested for all experiments as Suppl. Fig. S6. However, we prefer to show CFU/OD in the main figures. Under some conditions we are comparing strains/conditions with differences in the final OD. We feel that showing CFU/OD is more appropriate to show the “non-culturability”.

The authors should complement their *guaAB* deletion mutant.

Answer: *guaBA* complementation strains were generated and the results of CFU enumeration are included in Fig 2B.

On lines 78-80, the authors state that “These conditions mimicked amino acid starvation and resulted in significantly reduced culturability of the (p)ppGpp⁰ mutant (Fig. S1B).” However, statistical significance is not reported on the figure.

Answer: *Statistical significance is added to the figure*

On lines 102-103, the authors state that “Thus, deletion of *purR* in the (p)ppGpp⁰ mutant will likely lead to increased GTP levels.” This change in GTP was not confirmed using LC-MS/MS. Quantifying GM/D/TP levels in the mutant would strengthen this claim.

Answer: *We tried to measure the nucleotide levels in the purR mutants. Unfortunately, the results were not conclusive. Since the purR mutant results are not central to the manuscript, we now deleted the results obtained with this mutant (shown previously in Fig 2B). Thus, we do not have to speculate on the nucleotide pool in this strain. The role of PurR for nucleotide balance and antibiotic tolerance will be subject to further investigation.*

The authors state that loss of ppGpp results in accumulation of GTP, which is toxic to *S. aureus* in late stationary phase. Their LC-ms/ms data in Fig. 3D is supportive of this statement, and it is consistent with other reports in the field. Looking at Fig. 3D, it is possible that with GTP accumulation, GMP and GDP also accumulate, and these lower energy nucleotides are more abundant than GTP in the ppGpp⁰ strain. This suggests that the ppGpp⁰ cells are deenergized in late stationary phase, and perhaps their energy charge is <0.5, which renders them non- culturable/viable (PMID 4333317). Perhaps some of the defects observed late-stationary *S. aureus* cultures that lack ppGpp is linked to their overall energy charge and not just the accumulation of GTP.

Answer: *To address this interesting point, we calculated the energy charge of the cells. (p)ppGpp resulted in decrease of all guanine nucleotides GTP, GMP and GDP, thus the ratio between nucleotides was not significantly changed. We also calculated the energy charge based on ATP, ADP and AMP levels $(ATP) + 0.5 [ADP]/[ATP] + [ADP] + [AMP]$. The energy charge was indeed significantly decrease in (p)ppGpp⁰ mutants. The data are now included in Fig. S2. There is a significant decrease in the energy charge in the pppGpp⁰ mutants which might indeed contribute to loss of culturability. However, the low energy charge seems to be independent of GTP since it was also lower in the *guaAB*/pppGpp mutant which regained culturability. Thus, just changes in the energy charge can not explain our main findings. We now discuss this point:*

*“It has been proposed that intracellular energy is the main factor maintaining the culturability of E. coli on solid media 26,36. The (p)ppGpp mutant did indeed show a decrease in the energy charge which might contribute to VBNC formation. However, the p)ppGpp⁰ Δ*guaBA* double mutant showed similar decrease in ATP and energy charge but was still culturable. We therefore that the increased GTP level in (p)ppGpp⁰ is the major cause for VBNC formation. Furthermore, it has previously been shown that reducing only ATP in stationary phase *S. aureus* increases growth after antibiotic challenge 3. In *C. crescentus* fatty acid starved (p)ppGpp⁰ mutants also enter a VBNC like state 37. Lack of culturability was sought to be due to uncontrolled replication and thus the bacteria may be unable to recover from the miscoordination of cell cycle events. So far, we have no evidence for over-replication in the (p)ppGpp⁰ mutant. The*

transcriptional analysis even hinted to a down-regulation of some genes involved in DNA replication (e.g. polC) in the pppGpp0 mutant. Thus, possibly the combined reduction of energy and high GTP levels (resulting in lower membrane potential) may severely impair recovery after starvation. Our findings are consistent with previous observations in B. subtilis 14. The results already indicated that GTP levels or the GTP/ATP ratio, but not ATP levels, correlate with the ability to withstand amino acid limitation.”

Answer Reviewer 2:

Salzer et al. investigated the role of (p)ppGpp-mediated GTP levels in *S. aureus* and bacterial survival at stationary phase. They studied the culturability, and tolerance of starved cells. Although this has been studied in a previous study (Reference 18), the authors have brought in extra experiments to delve deeper into the mechanistic insights by looking at the involvement of the transcriptional start site of cytochrome aa3 system (qoxABCD) and the proton motive force of the cells. With the work done in this manuscript, the authors claim that (p)ppGpp is essential for maintaining GTP homeostasis, preservation of electron transport and PMF for culturability and antibiotic tolerance.

Overall, the main findings of the paper are interesting, supported by data, and it will be of interest to many readers in the field. Further data in addressing below questions/comments would support the conclusions of this study further, and link the study into the context of literature to influence thinking in the field.

Specific comments:

- 1) Line 102-104, The authors hypothesise that deletion of *purR* in (p)ppGpp null mutant should lead to increased GTP levels, and this is why the culturability of this strain decreases further. Have the authors considered checking the GTP levels of this *purR*/(p)ppgpp null mutant strain? Including the GTP levels of this strain in the manuscript would strengthen their statement.

Answer: *We tried to measure the nucleotide levels in the purR mutants. Unfortunately, the results were not conclusive. Since the purR mutant results are not central to the manuscript, we now deleted the results obtained with this mutant (shown previously in Fig 2B). Thus, we do not have to speculate on the nucleotide pool in this strain. The role of PurR for nucleotide balance and antibiotic tolerance will be subject to further investigation.*

- 2) Line 107, and materials and methods – What is the concentrations of guanine used in this work? Was this Guanine hydrochloride or what version of this purine nucleotide was used in this study? Was the pH of the media kept the same when this nucleotide was added or did you need to re-adjust the pH by adding NaOH? If so, was there an alteration to the salt levels within

the medium? This is lacking methodology details that are important for this to be reproducible by others.

Answer: *Guanine concentrations are not included in Fig 2 and in material and methods. The pH was buffered and did not change due to guanine addition.*

2) Line 206, The threshold used for log2 fold changes were >1 or <1 , should the cut off be not more stringent (i.e. above 1.5 or even 2?) I recommend >2 and <2 to be used instead of >1 , <1 .

Answer: *Thanks for this suggestion. Based on your recommendation we changed the threshold to log fold change <2 . This reduced the number of significantly regulated genes (115) and made the results indeed more meaningful. We included a **new figure 7A** based on data shown in a new sheet of data table S3.*

4) To further validate the RNA-Seq results, the authors should consider using some of the TCA-cycle mutants they refer to in their manuscript. They should be easily transducible into the (p)ppGpp null-background if their regulation is a cause for the GTP-dependent culturability. They claim that the downregulation of TCA cycle and respiration may account for the reduced membrane potential observed during stationary phase starvation. It would be ideal to prove this claim with further experiments utilising some of the TCA-cycle mutants. Additionally, their transcriptional start sites could also be analysed similar to that of qoxABCD studied in the manuscript.

Answer: *We transduced suggested mutations into our strains. However, the results are not very conclusive:*

- *the qox mutants showed severe alteration in growth. This made interpretation of the results regarding cultivability rather difficult.*

- The **sucA mutant** showed no difference in culturability indicating that the inhibition of the TCA cycle is not sufficient for the observed effects.

2023-04-11 CFU/OD sucA mutant

The more thorough analysis of the RNAseq data (using the higher threshold as recommended) made us doubt whether the expression of the TCA cycle genes are at all meaningful. Only minor changes for some genes were observed. Also, from previous analyses there is little evidence that (p)ppGpp has a direct role on TCA cycle genes. Due to lack of evidence, we now avoided the discussion concerning the role of the TCA cycle.

5) Figure S5, What is the OD-CFU ratio of WT, (p)ppGpp-null and additional mutants used in this study? If you are inoculating the same starting OD for the tolerance assays, does that translate to the same CFU starting?

Answer: Thanks for these thoughts and the link to this interesting article which nicely describes the main features of the stationary phase of *E. coli*. Some of the described features may also apply to *S. aureus*. In our main experiments we analyzed bacteria after 24h of growth which is likely not yet the deep stationary phase. The onset of the stationary plateau is clearly visible through OD measurements (e.g. see Fig1A).

When we inoculate the strains grown in CDM to the same initial OD we observed a severe delay in growth (longer lag) of the (p)ppGpp mutant compared to the wild type. That's why we used LB medium for preculture. In this medium (because of low GTP levels) the pppGpp0 was as culturable as the wild type.

To ascertain that the OD measurement indeed reflects bacterial numbers we performed additional microscopic analyses. In our growth conditions the OD value is highly concordant with the bacterial number enumerated by cell counting in the microscope. We now included the result of stationary phase wild type and (p)ppGpp mutant in the new Fig. 1C.

However, we prefer to show CFU/OD in the main figures. Under some conditions we are comparing strains/conditions with differences in the final OD. We feel that showing CFU/OD is more appropriate to show the "non-culturability". Nevertheless, we added now the CFU/ml values for the experiments as new Suppl. Fig 6. The overall outcome of the experiment remained the same.

6) Figure 9, This model is not fully supported by data. There are arrows to TCA cycle, however authors have not proven this. Also, directly linking (p)ppGpp to antibiotic tolerance is too far of

a stretch if only ciprofloxacin and oxacillin is tested. Please refine the model appropriately, or provide more work or references to support the arrows indicated.

Answer: *We have now rewritten the discussion and included a new model (see **new Fig. 9**)*

7) There have been links of c-di-AMP and ppGpp crosstalk in the literature. *S. aureus* c-di-AMP mutant survival has been linked to the qoxABCD system in a suppressor analysis, potassium transport, aerobic respiration and membrane potential. Authors do not discuss any of such literature in the context of their new findings, which could add value to the discussion of this manuscript and influence some thinking in Staphylococcus, and nucleotide signalling research fields.

Answer: *Thanks for this interesting link. We now discuss these findings in our discussion:*

“ Interestingly, the second messenger c-di-AMP is dispensable under anaerobic conditions but promote the survival under Qox dependent respiration 40. This could not be explained by protection from ROS alone. (p)ppGpp was shown to increase c-di-AMP synthesis possibly via the guanosine pathway 41,42. Thus, requirement for c-di-AMP and (p)ppGpp under aerobic growth may be functionally linked to proper Qox activity. This may help to maintain electron transport and allow the mutant to survive under aerobic conditions. Anaerobic respiration occurs independently of Qox via nitrate reductase NarGHJ. Interestingly, narGHJ expression is significantly upregulated in the (p)ppGpp0 mutant under high GTP conditions.”

9) Line 82, Why 17h and 24 h were chosen? How do you define mid-stationary? Have the authors looked at the culturability of the strains over a longer term than 24 hours?

Answer: *17h and 24 hours are approximately 6 and 13 hours after entry into stationary phase as illustrated Fig. The difference between wild type and (p)ppGpp mutant at later time points are even more prominent (see new Fig. S1)*

Answer to Reviewer 3:

In this manuscript, "(p)ppGpp-mediated GTP homeostasis ensures the survival and antibiotic tolerance of Staphylococcus aureus during starvation by preserving the protonmotive force", by Professor Wolz and colleagues showed that (p)ppGpp-dependent restriction of GTP pool contributes to the culturability of starved cells and antibiotic tolerance. The manuscript demonstrates a commendable level of clarity in its presentation. The key findings are very well supported by experiments. The presentation of the data and figures exhibits clarity. However, some areas need attention or clarification.

1) Lines 45-49 are difficult to follow, mainly when the concept of (p)ppGpp synthesis correlates inversely with the GTP pool is mentioned. It takes multiple readings to connect this information with the previous lines.

Answer: *Thanks for the comment. We have rewritten the sentences.*

2) In lines 75-76, please highlight the significance of the strains in the context of the study.

Answer: We now explain why we used these additional strains. Reproducibility in MRSA strain USA300 and phage-free strain SH1000 (effect not phage related):

“We next confirmed that reduced culturability in (p)pppGpp0 mutants is a general phenomenon. (p)ppGpp0 mutants of strain USA300 and SH1000 showed a similar reduction in culturability (Fig. 1D, Fig S6A). USA300 is a highly virulent methicillin-resistant strain. SH1000 is a Sigma factor B-positive and phage-cured derivative of HG001. The parental strain HG001 is naturally deficient in the activity of the alternative sigma factor B and carry three native prophages. Thus, decreased culturability when (p)ppGpp is lacking is independent of methicillin-resistance, sigma factor B activity or native phages.”

- 3) While lines 77-80, describe an experiment that was performed to induce the expression of Rel-dependent (p)ppGpp synthesis in (p)ppGpp^o mutant, line 69 reports that mutants were unable to synthesize (p)ppGpp. I have difficulty understanding the results of (p)ppGpp^o mutant. Also, supplementary S1B should have included the control group for both WT and mutant that has not been treated with mupirocin.

Answer: We have rewritten the statement for better understanding and added the control in new Fig S1: “ In *S. aureus* (p)ppGpp can be synthesised by three different synthetases (Rel, RelP or RelQ). Rel dependent stringent response can be initiated by mupirocin, a tRNA synthetase inhibitor mimicking amino acid starvation. Addition of mupirocin to exponentially growing bacteria resulted in a slight bactericidal effect in the wildtype. However, in the (p)ppGpp0 mutant significantly less CFU were recovered upon mupirocin challenge. Thus, (p)ppGpp also protects from stress exerted by amino acid starvation in the exponential growth phase (Fig. S1B) “.

- 4) I request authors to explain the results Figure S1C particularly focusing on why a significant change is observed at 7 hours but not at 24 hours.

Answer: We had some variation at the 24 h time point in our essay. Thus, the data did not reach significance. However, in our RNAseq analysis pms add rsaD are both significantly downregulated.

- 5) What could have been the possible reason for relP complementation to not show the effect?

Answer: We have no real explanation for this finding. Possibly relP is not active in stationary phase bacteria or we just did not reach significance in our essay.

- 6) Please mention the mutants in line 94, when they are first introduced.

Answer: Mutants are now mentioned.

- 7) Line 102-104, the authors concluded that deletion of purR resulted in elevated GTP levels that were detrimental to bacterial susceptibility. The authors did not measure the levels of GTP in this strain. How do they conclude that it is because of GTP and not ATP?

Answer: We tried to measure the nucleotide levels in the purR mutants. Unfortunately, the results were not conclusive. Since the purR mutant results are not central to the manuscript, we now deleted the results obtained with this mutant (shown previously in Fig 2B). Thus, we do not have to speculate on the nucleotide pool in this strain. The role of PurR for nucleotide balance and antibiotic tolerance will be subject to further investigation.

8) In Figure 2C, why no significant change was observed in WT vs (p)ppGpp^o mutant at 10µg/ml guanine concentration?

Answer: *We assume that the concentration is just too low to result in increased GTP levels in the (p)ppGpp^o mutant.*

9) Please mention about CCCP control and its role in line 127.

Answer: *CCCP was used as negative control (RSG assay measures bacterial reductase activity which in turn is a marker for electron transport chain activity → CCCP disrupts electron transport chain function)*

10) I would appreciate it if author could explain why most of the experiments did not involve the rels^{yn-} or relQ-complemented strains.

Answer: *The experiments were first contacted with the triple-mutant, which is not easy to complement. Thus, the complementation experiment was only performed at a later stage of the project. Also, the complementation with the single enzymes resulted only in partial complementation.*

11) Please include the level of statistical significance between ΔguaBA and (p)ppGpp^o ΔguaBA in figure 3E and between WT and (p)ppGpp^o in figure 8D.

Answer: *Statistics are added*

12) Please mention the time point at which the experiment in figure 4b and 4c was conducted.

Answer: *Chloramphenicol treatment was done at 7.5 or 9h (need to check in the lab book again), incubation with CAM for 30 min. This is now added in the legend.*

13) The significance of the current study is not adequately elucidated in the manuscript.

Answer: *We have rewritten the discussion to emphasize the significance of the study. We also included a new model (new Fig. 9)*

Rebuttal revision

Thanks again for the careful reading of our manuscript.

Reviewer 1:

Line 81: please change (p)pppGpp to (p)ppGpp.

Line 327: please add the bracket “(“ before (p)ppGpp.

Line 331: please add a hyphen between “VBNC” and “like.”

Line 353: please italicize “*S. aureus*.”

Line 430: how long were the biofilms cultured?

Answer: We have corrected all the errors found accordingly

Reviewer 3:

Please introduce (p)ppGpp^o in line 68 instead of line 70.

2. In line 81, authors mentioned general phenomena, I believe they intended to convey it is not a strain-specific phenomenon. Please revise the sentence.

3. In line 118, Fig 3E is mentioned, but there is no Fig 3E in the panel.

4. In line 121 and 125, Fig 2D is referenced; I believe the authors intended to refer to Fig 2C instead. Similarly, in line 129, fig 2C should be corrected to Fig 2D.

5. The scale used in Fig 1B and 1E differs in representation. Please correct for uniformity across figures.

6. There are several typographical errors in the manuscript, like line 345-PMN, line 327-(p)ppGPP^o, line 353-italicized *S. aureus*

Answer: We have corrected all the errors found accordingly and have provided a new figure 1